# Oceanic regions shape the composition of the Antarctic plastisphere

Ana L. Lacerda [1,2] ✉, Maíra C. Proietti [1,3], Felipe Kessler [4], Carlos R. Mendes [1],
Eduardo R. Secchi [1] & Joe D. Taylor [5] ✉

Antarctica, once considered pristine, is increasingly threatened by plastic pollution, with debris found in its waters, sediments, sea ice, and biota. Here, we provide a comprehensive molecular survey of both prokaryotic and eukaryotic diversity on plastics around the Antarctic Peninsula, addressing a gap in existing research. Using eDNA metabarcoding, we identified diverse communities, with Pseudomonadota and Bacteroidota dominating prokaryotic communities, while Gyrista (mostly diatoms), Fungi and Arthropods were prevalent among eukaryotes. Geographic location significantly influenced community composition, with differences between the Bransfield Strait and the Gerlache Strait/Bellingshausen Sea. Polymer type and plastic shape did not impact species richness or community structure. These findings offer new insights into the complexity of the Antarctic plastisphere, highlighting potential impacts on biodiversity, ecosystem functions, and the broader implications of marine plastic pollution.

Antarctica was once considered a pristine environment, but several studies have now highlighted that this region contains a range of pollutants, including plastic pollution in surface waters[1–3], deep-sea sediments[4,5], sea ice[6], and biota[7]. A 30-year monitoring study of debris in the Southern Ocean identified variable trends in debris concentration over time, and plastics represented more than 80% of debris items in two locations of the Scotia Sea[8]. The interaction of plastics with Antarctic wildlife has been increasingly reported and includes ingestion by a range of species, from small benthic animals to megafauna[9,10], as well as entanglement, recorded for several marine mammal species[7,11,12]. As plastic production and use are steadily increasing, so will the concentration and impacts of this type of pollution on the environment.

One characteristic of plastics in the ocean is their ability to host and transport organisms among regions, which can potentially result in species invasions[13], including in the Antarctic Peninsula[14]. In marine systems, microbes quickly colonise plastics[15], creating the "plastisphere"[16]. As a consequence, ecological successions may occur, leading to mature communities[17,18] that can be composed of a wide range of prokaryotes[19–21] and eukaryotes[22,23]. The term "plastisphere", initially used to describe microbial communities associated with plastics in marine systems, has been expanded and now describes all organisms that live attached to plastics in aquatic and terrestrial environments[24,25]. Biofouling on plastics can influence their weathering and contaminant

absorbance[26], as well as their vertical transport through the water column[27].

The plastic-associated communities have been known for some time in Antarctic marine waters. In the early 2000s, an assemblage of animals attached to a piece of plastic that had washed ashore on Adelaide Island was documented[28], with at least ten species spanning five different phyla. In addition, our research group has evaluated fungi in biofilms from plastics collected in the Western Antarctic Peninsula[2], using eDNA metabarcoding[29]. The structure and function of prokaryotes in the Antarctic plastisphere have been described based on only two plastic items, one from land and one from the sea[30]. Two further recent studies have looked at the colonisation of plastics by microorganisms in the Ross Sea[31] and in microcosm experiments on Livington Island, in the South Shetlands archipelago[32], both using molecular techniques.

Although plastic pollution research is increasing in Antarctica[33,34], there have, to our knowledge, been no wide-scale molecular surveys to describe the diversity of both prokaryotes and eukaryotes of the plastisphere from plastics floating for unknown periods of time in the Southern Ocean. The long-term monitoring studies developed by the High Latitude Oceanography Group—Grupo de Oceanografia de Altas Latitudes (GOAL, in Portuguese) in the last 20 years have shown increasing ice melting and changes in phytoplankton communities at the Antarctic Peninsula, mostly attributed to global warming[35–37]. It is, therefore, important to gain baseline

¹Instituto de Oceanografia, Universidade Federal do Rio Grande (FURG), Rio Grande, Brazil. ²Littoral Environnement et Sociétés (LIENSs), UMR 7266, CNRS, La Rochelle Université, La Rochelle, France. ³The Ocean Cleanup, Rotterdam, The Netherlands. ⁴Escola de Química e Alimentos, Universidade Federal do Rio Grande (FURG), Rio Grande, Brazil. ⁵UK Centre for Ecology and Hydrology, Benson Lane, OX10 8BB Wallingford, UK. ✉e-mail: ana.de_figueiredo_lacerda@univ-lr.fr; joetay@ceh.ac.uk

data on the composition of the plastisphere so we can better understand potential future changes in response to anthropogenic and climate impacts. To continue improving the understanding of life associated with plastics in Antarctica, here we used a multi-marker eDNA metabarcoding approach to characterise the plastisphere in the Antarctic Peninsula. This study investigated how these communities varied in different plastic shapes, polymeric composition and regions (northwestern and southwestern Antarctic Peninsula), and discussed their ecological role in the Southern Ocean.

## Material and methods

### Sampling area

Plastics were sampled at the sea surface (water-air interface, around 15 cm depth) of 12 sampling stations during the XXXVI Antarctic Operation and 7th expedition of project "Biological Interactions in Marine Ecosystems off the Antarctic Peninsula Under Different Impacts of Climate Change—INTER-BIOTA", in 2017 (Fig. 1). Samples were collected between 61° and 64°S using a manta net with a 100 × 21 cm mouth and a 330 μm mesh. At each station, the net was deployed from the windward side of the vessel via a large A-frame and trawled at a speed of 2.5–3.5 knots for 15–55 min, depending on weather and logistical conditions. The study area covered the Gerlache Strait, which separates the Anvers and Brabant Islands from the Antarctic Peninsula; the Bransfield Strait, between the southern Shetland Islands and the Peninsula; and the northeastern Bellingshausen Sea. Salinity and sea temperature were recorded concurrently with plastic sampling at each site. HPLC-derived measurements of chlorophyll-a concentrations were obtained from Ferreira et al.[35], following the methodology established by GOAL. Detailed information on the environmental parameters are provided as additional data. (Fig-Share: https://doi.org/10.6084/m9.figshare.28784807.v1).

To better evaluate the effects of oceanographic features on the plastisphere composition, the sampling stations were split into two data sets: the Northwestern (NW) region, which includes the Bransfield Strait, and the Southwestern (SW) region, which comprises the Gerlache Strait and the region under strong influence of waters advected from the Bellingshausen Sea. These two marine regions present different bathymetry (Fig. 1) and sea-ice coverage, which are the major controllers of the biogeochemical spatial variability in the Southern Ocean, making them distinct biogeochemical regions[38]. At sub-surface and deeper levels, the study region is influenced by water masses flowing from the Bellingshausen and Weddell Seas,

respectively called Transitional Bellingshausen Water (TBW) and Transitional Weddell Water[39]. The SW region, more influenced by the TBW, is characterised by a regime of warmer, more nutrient-rich and more productive waters, while the NW region has a greater influence of cold, more saline waters derived mainly from the Weddell Sea and generally contains lower nutrient concentrations[40].

### Characterisation of plastics

After each trawl, the contents of the collection cup were placed in an aluminium bag and frozen at −20 °C for posterior analysis of the plastisphere. In the laboratory, samples were thawed separately and placed in a sterile container filled with artificial sterile salt water (salinity 35) for manual separation of floating plastics (higher than 1 mm in length) and the organic matter (i.e., the zooplankton and macroalgae that were sampled along with plastics)[41]. Plastic pieces were picked up using sterile forceps, measured over their total length and classified into categories according to their size (microplastic: <5 mm, and mesoplastic: 5–200 mm[42]; shape (fragment, foam, line, pellet and film)[43], and polymer composition[2]. Each plastic piece was placed individually in a microcentrifuge tube with absolute ethanol (reagent grade, MERK) to preserve the DNA of the associated organisms, and 32 were submitted to genetic analysis (present study). The polymer composition of 28 out of the 32 samples used for DNA analysis was determined through Fourier Transform Infrared Spectroscopy (FTIR) with a SHIMADZU spectrometer, model Prestige 21, using a diffuse reflectance module, 24 scans and 4 cm$^{-1}$ resolution. FTIR procedures and data analysis follow the standard practice ASTM E1252-98 (2013)[44].

### Scanning electron microscopy (SEM)

Fourteen additional plastic pieces (different from the ones submitted to DNA extraction) were observed through scanning electron microscopy (SEM) for the detection of biofilms, aiming to investigate the morphology of the plastisphere organisms. Plastics were initially dehydrated in absolute ethanol (Reagent-grade, MERK), followed by fixation to an aluminium sheet with carbon tape and coated with a 20–30 nm gold layer. The biofilm was observed using a JEOL microscope (JSM 6610LV, JEOL, Tokyo), operated at 10–20 kV at a working distance of 10–26 mm. Each item was imaged at different magnifications (20× to 40,000×) to better identify the diversity of organisms.

The main groups found on plastics were qualitatively described.

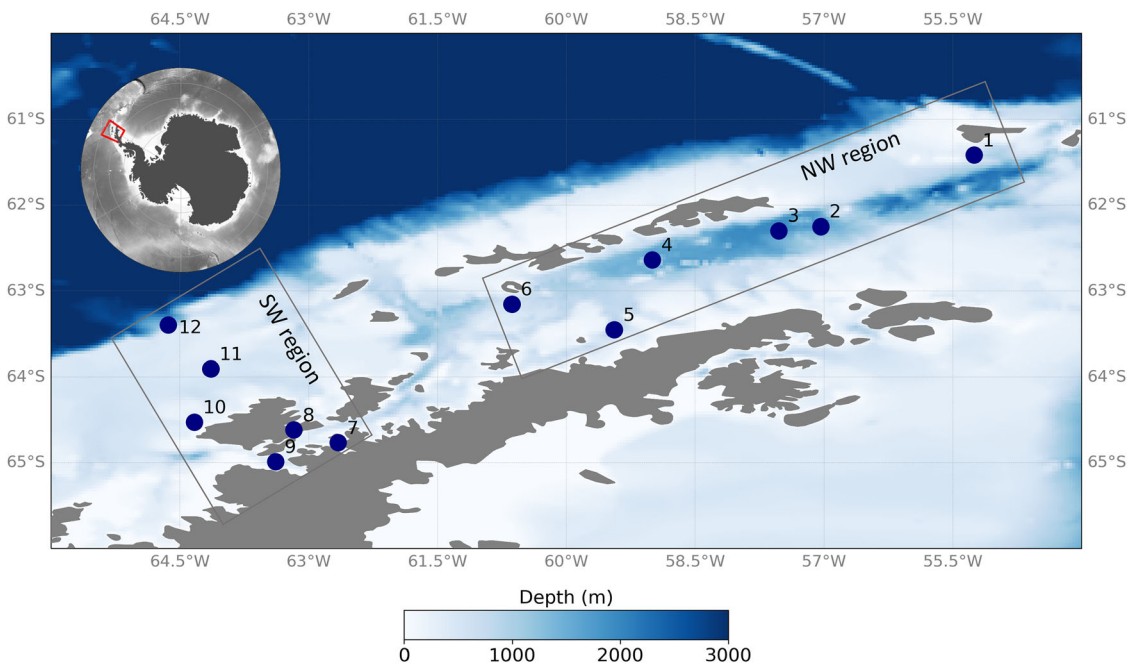

**Fig. 1 | Sampling area of floating plastics and their associated plastisphere in Antarctica.** Sampling sites 1–6: Northwestern region (NW); Sampling sites 7–12: Southwestern region (SW). This map was created with Python software by using the "*matplotlib*" package.

### eDNA extraction, amplification and sequencing

Plastic pieces were rinsed with sterile seawater before DNA extraction to remove organisms that co-occurred with plastics during sampling. The total DNA of plastic biofilms was extracted using a PowerSoil DNA extraction kit (Qiagen)[45], with some modifications from the manufacturers' instructions, as described in Lacerda et al.[29]. The quality and concentration of extracted DNA were checked by spectrophotometry using Biodrop DUO (Harvard Bioscience™). We then PCR amplified the V4 region of the 16S rRNA gene to target prokaryotes (Forward primer 515 f 5′-GTGYCAGCMGCCGCGGT AA-3′, and reverse primer 806r 5′-GGACTACNVGGGTWTCTAAT-3′)[46], the 18S rRNA gene V4 region (Forward primer TAReuk454 5′-CAGCASC YGCGGTAATTCC-3′, reverse primer TAReukRev3 5′-ACTTTCGTTCT TGATYRA-3′)[47], and the 18S rRNA gene V9 region (Forward primer 1391 f 5′-GTACACACCGCCCGTC-3′, reverse primer EukB 5′-TGATCCTTC TGCAGGTTCACCTAC-3′)[48]. Two regions of the 18S rRNA gene were used due to their different resolutions on the diversity of eukaryotes[49]; this was confirmed in a recent study on the plastisphere from the South Atlantic Ocean[50]. PCR reactions and conditions for all molecular markers are detailed in Lacerda et al.[29,50]. Library preparation and sequencing were performed using a 2x 300 bp V3 sequencing kit on an Illumina Miseq[29].

### DNA sequence processing and data analysis

Primer sequences were removed using Cutadapt (version 1.8)[51]. Raw paired-end Illumina reads were processed using the DADA2 pipeline (v1.8) in *R* environment[52] to generate amplicon sequence variants (ASVs) from 16S and 18S rRNA gene sequences. Quality filtering and trimming were performed using the filterAndTrim() function, with region-specific parameters to remove low-quality reads and sequencing artefacts. For the 16S, reads were truncated at 220 bp (forward) and 200 bp (reverse), using maxN = 0, maxEE = c(2,2), truncQ = 2. For 18S-V4, truncation lengths were set to 230 bp (forward) and 200 bp (reverse), with the same filtering thresholds. For 18S-V9, both forward and reverse reads were truncated at 100 bp, with maxEE = c(3,3) to accommodate the shorter amplicon while retaining the same quality parameters. Following filtering, dereplication was performed separately for forward and reverse reads using the derepFastq() function, and error rates were learned using learnErrors(). Reads were then denoised with the dada() function, followed by the merging of paired reads with mergePairs(). Chimeric sequences were removed using the removeBimeraDenovo() function in consensus mode, and a final ASV table was constructed using makeSequenceTable(). Taxonomic assignment was performed using the assignTaxonomy() function with the naïve Bayesian classifier (Wang et al. 2007) and a minimum bootstrap confidence threshold of 80. Prokaryotic (16S) sequences were classified against the SILVA v138.2 database[53], while eukaryotic (18S rRNA) sequences were classified using the PR² v5.0.1 database[54].

In the 16S rRNA dataset, eukaryotes, mitochondria, and chloroplasts were excluded before further analysis. Similarly, in both 18S rRNA datasets, unknown domains and certain large metazoans (Salpida and Craniata) were manually removed during analysis, as they were unlikely to be associated directly with the plastics. Additionally, for confirmation of taxonomy, sequences underwent verification via the "Basic Local Alignment Search Tool" against the comprehensive National Centre for Biotechnology Information (NCBI) Genbank database. A review of relevant literature provided information on the functional potential of prokaryotic communities. Rarefaction was performed prior to statistical analysis, and ASV tables were rarefied to 894 reads for the 16S rRNA marker, 476 reads for 18-V4, and 997 reads for 18S-V9 markers.

Differences in alpha and beta diversity of plastic-associated organisms (ASVs richness and community structure) among plastic categories (size, shape and polymer composition), as well as by region, were evaluated. An analysis of variance (ANOVA) was performed to check for differences in ASV richness per plastic category and region. To assess beta diversity, we normalised ASV read counts by converting them into relative abundance (RA) values. The beta diversity was measured as the average distance from the individual plastic to the category's median using Bray-Curtis for the 16S rRNA dataset, and the binary Jaccard index for 18S rRNA V4 and V9 datasets. For eukaryotes, we used unweighted Jaccard matrices instead of Bray-Curtis distances to focus on presence rather than abundance. This approach accounts for the considerable variation in 18S rRNA gene copy numbers among different eukaryotic species, as well as differences in RA between single-cellular and multicellular organisms[55].

To verify if differences in community structure could derive from within-group variations, multivariate homogeneity tests of group dispersions (PERMDISP) were conducted. Furthermore, a permutational multivariate analysis of variance (PERMANOVA), with fixed factors and 9999 permutations, was employed to assess potential disparities in beta diversity among categories. All statistical analyses were carried out using the vegan package[56] within *R* studio 1.1.456, and differences with $p \leq 0.05$ were deemed statistically significant. Principal coordinate analysis was performed using the ggplot2 package[57] to verify differences in the community composition according to regions and plastic categories based on either Bray-Curtis dissimilarity (prokaryotes) or Jaccard distance matrices (eukaryotes). Shared and unique ASVs according to region and plastic categories for the 16S and 18S rRNA datasets were visualised by Venn diagrams created online at the InteractiVenn website[58].

## Results

### Environmental characterisation of the studied regions

Warmer surface waters were observed in the Gerlache Strait (SW region), with temperatures ranging from 1.56 to 3.23 °C. In contrast, the Bransfield Strait (NW region) exhibited colder surface waters, with temperatures ranging from 1.03 °C to 2.48 °C (See additional FigShare data: https://doi.org/10.6084/m9.figshare.28784807.v1). Regarding salinity, the NW region displayed a range from 33.9 to 34.2, while the SW region showed a variation ranging from 33.5 to 33.9. In terms of chlorophyll-a concentrations, the NW region had lower values, ranging from 0.19 to 1.14 mg.m$^{-3}$ (mean 0.52, SE ± 0.17), whereas the SW region exhibited higher concentrations, ranging from 0.26 to 4.65 mg.m$^{-3}$ (mean 2.10, SE ± 0.71).

### Morphology of plastisphere organisms

SEM revealed a number of different organisms living on floating plastics in Antarctica. We observed diatoms, fungi and bacteria attached to plastic fragments (Fig. 2A, D, F), lines (Fig. 2B, E) and foam (Fig. 2C). Since the preservation method of samples was focused on preserving the DNA (immediately frozen, followed by immersion in absolute ethanol), some cells/structures could have broken, but it was still possible to identify several groups living attached to plastics, reinforcing the presence of taxa detected with the molecular data.

### DNA sequence metrics

After quality filtering, the 16S rRNA gene dataset contained 321,829 reads from 21 successful samples (ten from the NW and 11 from the SW regions), comprising 1618 ASVs. Within the 16S rRNA dataset, the number of ASVs per sample ranged from 30 to 384, and the number of reads ranged from 885 to 55,923. For the eukaryotic markers, 23 samples had the 18S-V4 region successfully amplified (12 from NW, 11 from SW), with 393,082 reads comprising 676 ASVs, whereas 26 samples had the 18S-V9 marker amplified (14 from NW, 12 from SW), with 389,672 reads comprising 527 ASVs. The number of ASVs per sample ranged from ten to 147 and from nine to 109, while the number of reads ranged from 479 to 63,520, and from 1156 to 89,941 in the 18S-V4 and 18S-V9 datasets, respectively.

### Prokaryotic diversity in the Antarctic plastisphere

Within the 16S rRNA dataset, we detected 43 phyla of Bacteria and three Archaea phyla, as well as a number of "unclassified bacteria" (Fig. 3). The most abundant phyla within the dataset were Pseudomonadota (RA 44%) and Bacteroidota (RA 25%). Among Pseudomonadota (formerly Proteobacteria), the class Gammaproteobacteria was composed of 323 ASVs, representing 23% of the total RA, while Alphaproteobacteria contained 153 ASVs that consisted of 21% of the total RA of prokaryotes.

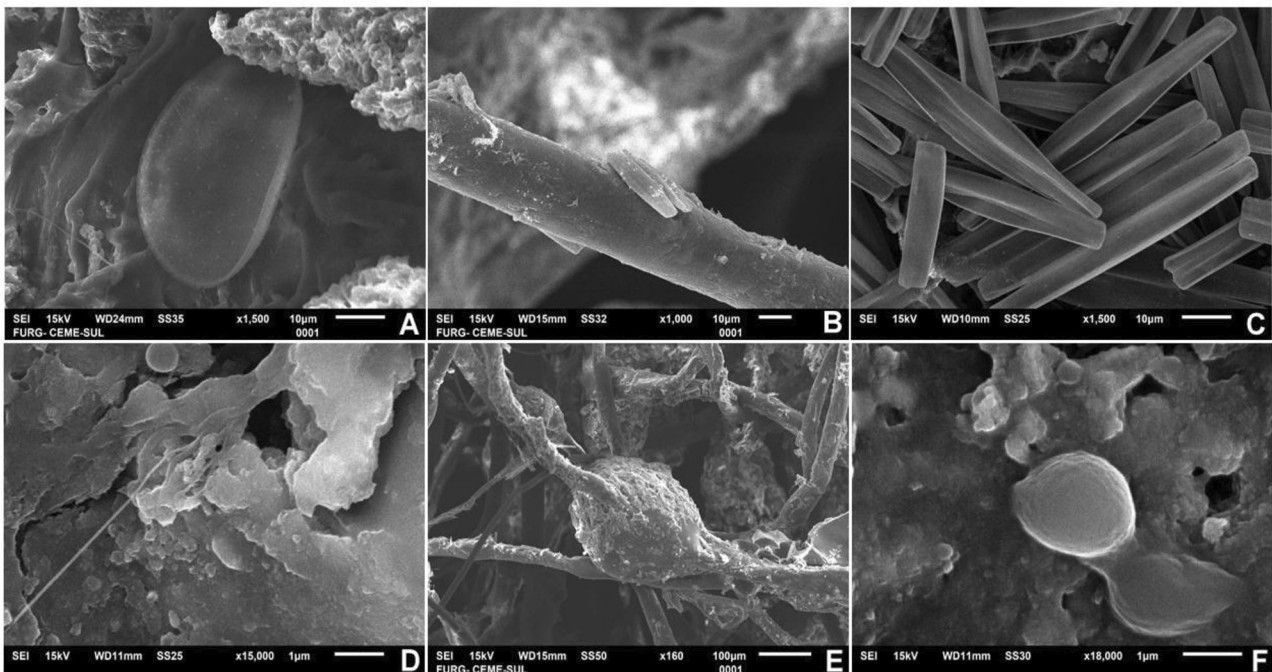

**Fig. 2 | Morphology of plastisphere organisms in Antarctica.** Scanning electron microscopy of organisms attached to floating plastics from the Antarctic Peninsula. Centric (**A**) and pennate (**B** and **C**) diatoms; fungi (**D** and **E**) and bacterial cells (**F**).

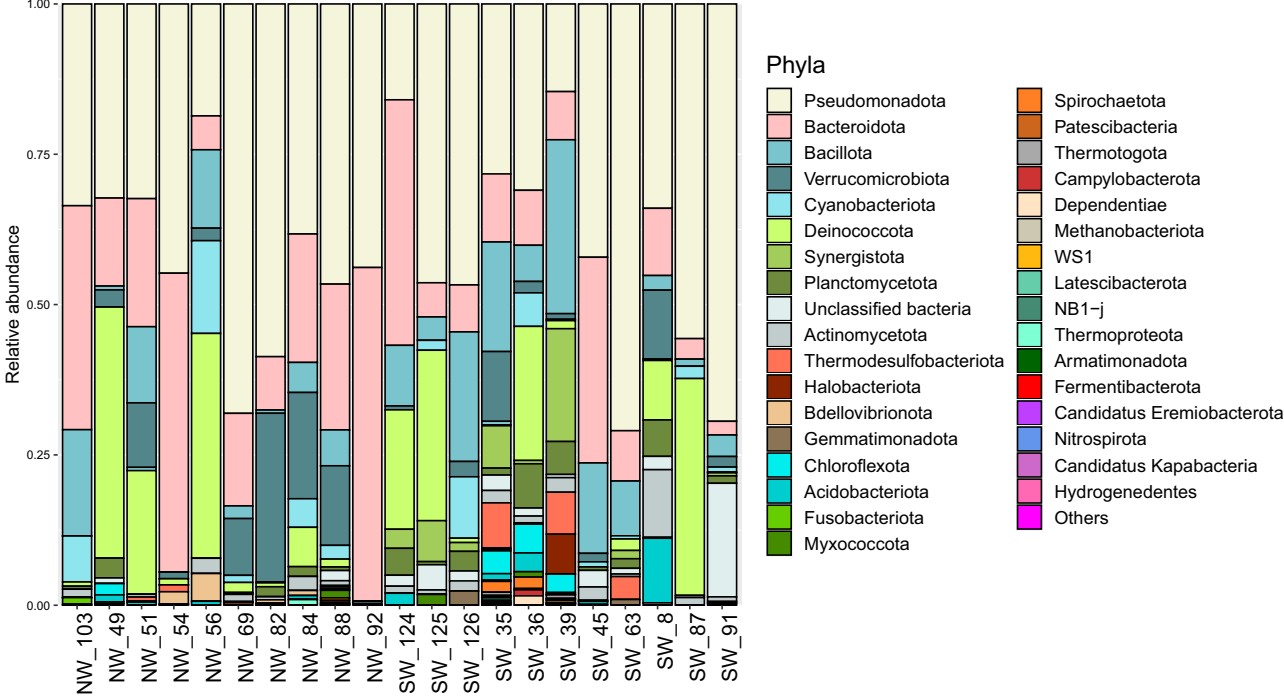

**Fig. 3 | Relative abundance of prokaryotes.** Prokaryotic phyla per plastic sample identified through the 16S rRNA gene according to the region (NW: Northwestern; SW: Southwestern) of the Antarctic Peninsula. "Others" comprises phyla with relative abundance lower than 0.005% (Chlamydiota, Aquificota, SAR324 clade - Marine group B, FW113, Fibrobacterota, LCP-89, Sumerlaeota, Zixibacteria, Cloacimonadota, Deferribacterota, Methylomirabilota, Thermoplasmatota and Caldisericota).

At the family level, the most abundant bacterial group was Flavobacteriaceae (Bacteroidota, with 63 ASVs and RA of 17%), and Paracoccaceae (Alphaproteobacteria_Rhodobacterales, with 93 ASVs and RA of 13%). The Archaea kingdom was composed of 15 ASVs classified into three phyla, Methanobacteriota, Thermoplasmatota and Halobacteriota, representing together an RA of 0.5%. The most abundant archaea ASV was *Methanosarcina* sp. (RA 0.4%, Frequency of occurrence (FO) 5%), whereas *Methanocorpusculum* sp., the second most abundant archaea ASV, occurred in more samples, but in lower abundances (FO 10%, RA 0.05%).

The most abundant prokaryotic ASV in the 16S dataset was *Marivibrio* sp. (RA 3.8%), and the second was *Bacteroides* sp. (RA 3.7%); both ASVs matched with higher percentage identities with uncultured environmental

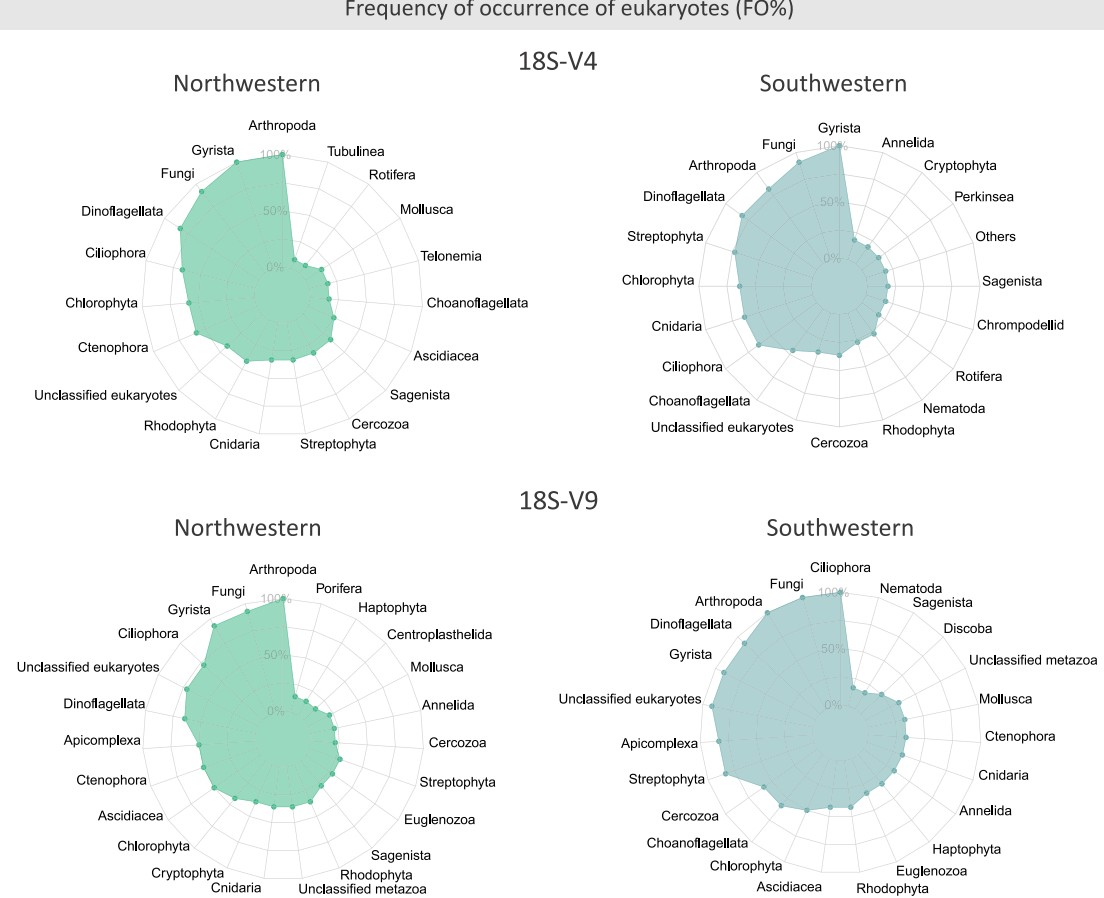

**Fig. 4 | Eukaryotes in the Antarctic marine plastisphere.** Percentage of eukaryotic frequency of occurrence (FO) of groups found in the marine Antarctic plastisphere, identified through the 18S rRNA gene (V4 and V9), separated by region (Northwestern and Southwestern). For better visualisation, only groups with FO higher than 10% are shown for the 18S-V4 and the 18S-V9 in the Southwestern, and for the 18S-V9 Northwestern regions.

sequences from NCBI. These two ASVs were present in almost half of the samples, with FOs of 43% and 48%, respectively.

In addition, we identified *Octadecabacter*, *Acinetobacter*, *Dokdonia* and *Articiflavibacter* species among the ten most abundant prokaryotic ASVs. Moreover, two *Polaribacter* species were also among the ten most abundant ASVs, one being classified as uncultured *Polaribacter* in NCBI (RA 1.6%, and FO 57%) and the other matching 100% with *Polaribacter sejongensis* (RA 1.3%, and FO 29%) collected from Antarctic soil (NCBI access number: NR_109324.1).

**Eukaryotic diversity in the Antarctic plastisphere**
The two eukaryotic datasets detected different taxonomic groups. For instance, Gastrotricha, Ichthyophonida, Perkinsea, Platyhelminthes, Radiolaria, Rotifera, Telonemia, unclassified Opisthokonta and Teleostei were identified only by the 18S-V4, whereas 12 other groups were exclusive to the 18S-V9 dataset: Amoebozoa, Bryozoa, Centroplasthelida, Discoba, Euglenozoa, Evosea, Excavata, Haptophyta, Myxozoa, Nebulidia, Picozoa and Porifera, and "unclassified metazoa". Twenty-three other eukaryotic groups were identified by both molecular markers: Annelida, Apicomplexa, Arthropoda, Ascidiacea, Cercozoa, Chlorophyta, Choanoflagellata, Chrompodellids, Ciliophora, Cnidaria, Cryptophyta, Ctenophora, Dinoflagellata, Discosea, Echinodermata, Fungi, Gyrista, Mollusca, Nematoda, Rhodophyta, Sagenista, Streptophyta and Tubulinea, along with "unidentified eukaryotes" (Fig. 4; See additional data, FigShare: https://doi.org/10.6084/m9.figshare.28784807.v1).

In both 18S datasets, only seven groups of eukaryotes presented FO higher than 50%. Arthropoda (primarily crustaceans), Gyrista (mainly diatoms), Fungi, Dinoflagellata, Ciliophora, and Streptophyta were consistently present (FOs between 52% and 96%) in both the 18S-V4 and 18S-V9 datasets, although their frequencies varied depending on the marker. For example, Arthropoda was the only group with FO 100% in the 18S-V9 dataset, while Gyrista was the only group with FO 100% in the 18S-V4 dataset. In addition to these groups, Chlorophyta was more abundant on the 18S-V4 (FO 61%), while Apicomplexa (FO 81%) was prevalent in samples identified with the 18SV9 rRNA. Many diatom ASVs matched species previously described in the NCBI for the Southern Ocean or polar regions, such as the benthic diatom *Navicula glaciei* (Genbank access number EF106789.1), *Chaetoceros socialis* (KX253957.1), *Corethron inerme* (AJ535180.1) and *Porosira glacialis* (ON888453.1). The 18S-V9 marker dataset showed many "unidentified/uncultured" eukaryotes (FO 81%), whereas this group was only observed in 39% of samples within the 18S-V4 dataset.

When considering the most frequent eukaryotic ASVs, the dinoflagellate *Cladocopium* sp. (formerly known as *Symbiodinium* sp.), the crustacean *Carpilius* sp., as well as an unidentified Ctenophora and the diatom *Grammonema* sp. (formerly known as *Fragilaria*) stood out in the 18S-V4 dataset. Meanwhile, for the 18S-V9 dataset, fungi species from genera *Aspergillus* and *Sterigmatomyces*, and the pennate diatom *Synedra* (Fragilariaceae family) were the most frequent ASVs. We also identified microeukaryotes described as animal parasites, such as the parasitic alveolate *Cephaloidophora* sp., and ciliate species belonging to the *Epistylis* and *Vorticellides* genera. Potential harmful fungi associated with plastics in Antarctica are not reported here, since they were previously described[29].

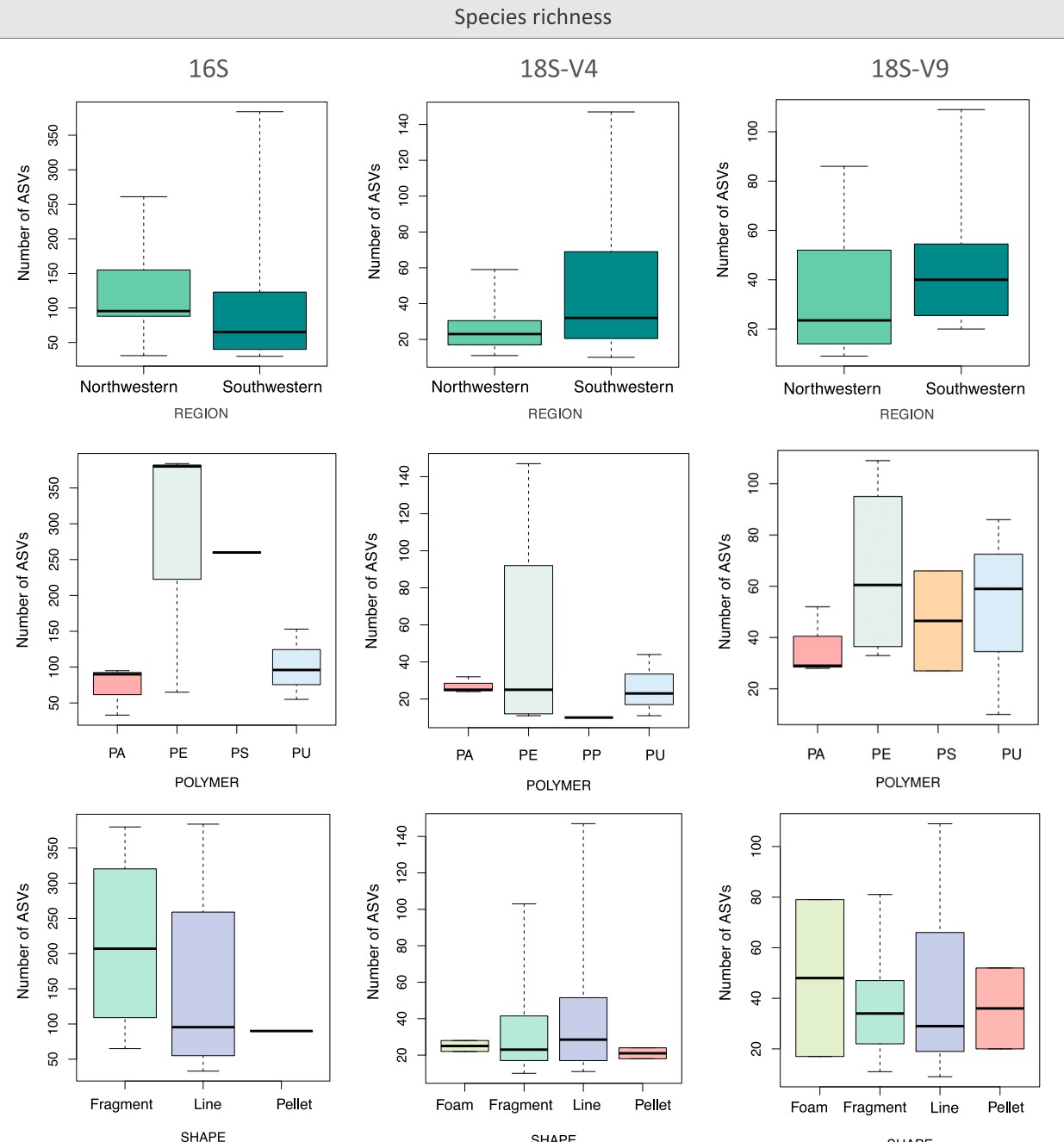

**Fig. 5 | Species richness in the Antarctic plastisphere.** Mean number of observed ASVs per plastic sample from the Antarctic Peninsula, according to geographic region (Northwestern $n = 14$, and Southwestern $n = 12$), polymer type (PA – polyamide $n = 3$; PE – polyethylene $n = 4$; PP – polypropylene $n = 1$*; PS – polystyrene $n = 2$; PU – polyurethane $n = 3$) and plastic shape (foam $n = 2$; fragment $n = 13$; line $n = 9$ and pellet $n = 2$). *Not considered for statistical analyses.

## Species richness and community structure according to region and plastic categories

The prokaryotic species richness (number of ASVs) was not significantly variable according to region or plastic categories (shape and polymer) (ANOVA, $p$ values > 0.05) (Fig. 5). However, the community structure of prokaryotes was different between plastics from the NW and SW regions (PERMANOVA; pseudo-$F = 1.4467$, $p = 0.03$), although it did not vary among any of the different plastic categories. Pseudomonadota, Bacteroidota, Verrucomicrobiota, Bacillota and Cyanobacteriota were highly abundant in both locations, but in the NW region, Deinococcota and Bdellovibrionota were more abundant, while in the SW region Synergistota, Planctomycetota, and unclassified bacteria were the most representative phyla.

For eukaryotes, ASVs richness did not vary according to any category of plastics or geographic regions, but as for prokaryotes, the community structure was different between regions for both 18S datasets (18S-V4: pseudo-$F = 1.3457$, $p = 0.001$; 18S-V9: pseudo-$F = 1.541$, $p < 0.001$) (Fig. 6A). No significant variations in community structure were observed based on plastic shape (18S-V4: pseudo-$F = 0.9815$, $p = 0.624$; 18S-V9: pseudo-$F = 1.0025$, $p = 0.448$) or polymer type (18S-V4: pseudo-$F = 0.9877$, $p = 0.522$; 18S-V9: pseudo-$F = 0.9945$, $p = 0.487$).

For eukaryotes identified using the 18S-V4 marker, fewer taxonomic groups were detected in the NW region compared to the SW region. Specifically, Nematoda, Chrompodellids, unidentified Opisthokonta, Gastrotricha, Radiolaria, Discosea, Perkinsea, Cryptophyta, Ichthyophonida, Apicomplexa, Annelida, Echinodermata and Platyhelminthes were absent

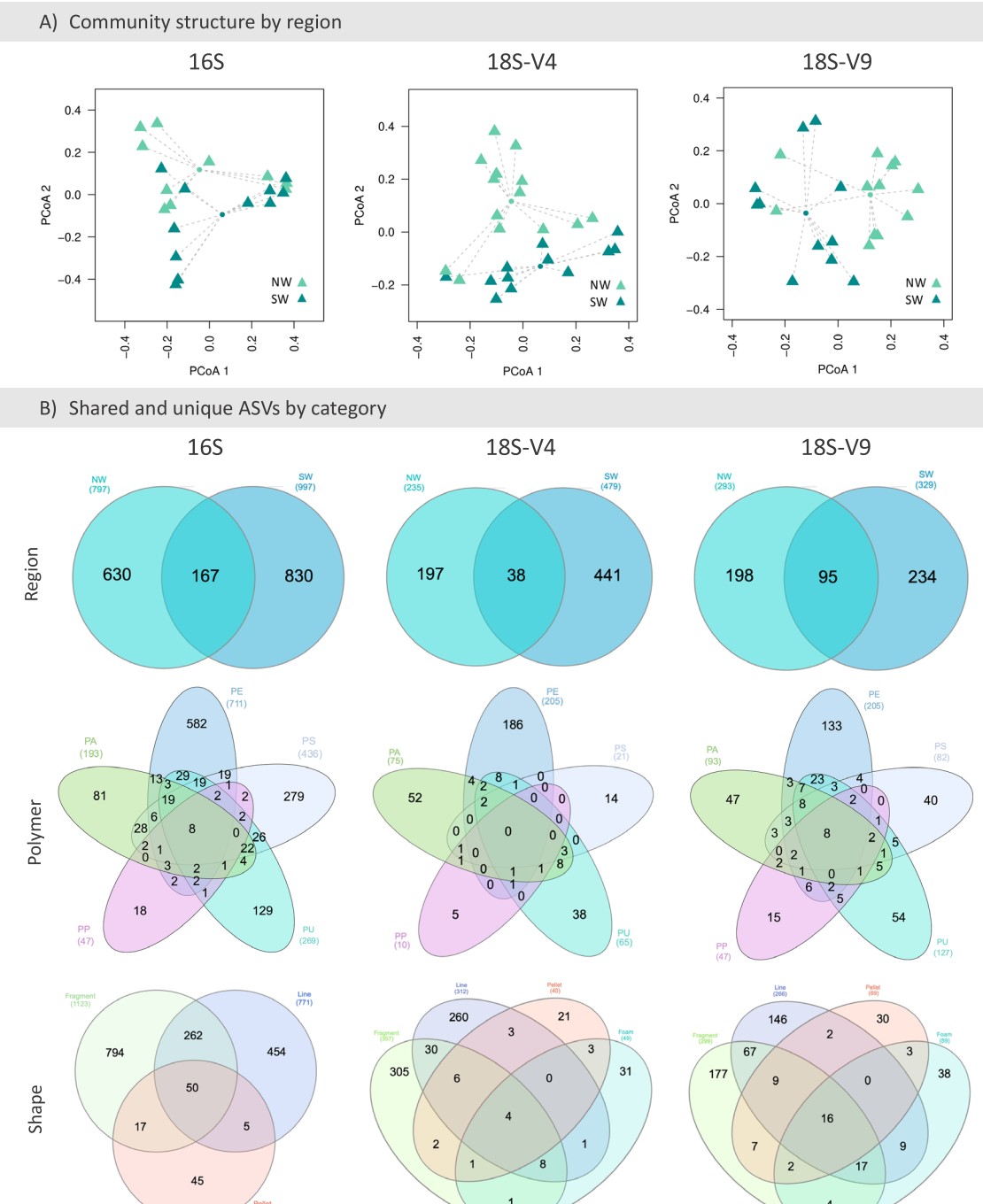

**Fig. 6 | Community structure of prokaryotes and eukaryotes. A** Non-metric multidimensional scaling (NMDs) plots of the marine plastisphere from the Antarctic Peninsula based on the Bray-Curtis distance matrix for the rRNA 16S, and on the Jaccard distance matrix for the 18S-V4 and 18S-V9 rRNA datasets, according to region (NW: Northwestern and SW: Southwestern). **B** Shared and unique ASVs between regions (NW and SW), polymer type (PA - polyamide; PE - polyethylene; PP - polypropylene; PS - polystyrene; PU - polyurethane) and plastic shape (foam, fragment, line and pellet).

from the NW region. In contrast, the SW region contained all eukaryotic groups identified with this marker, except for Ascidiacea. Among the eukaryotic groups detected using the 18S-V9 marker, a total of 31 groups were commonly observed in both the NW and SW regions, although their frequencies of occurrence varied by region. However, some groups were region-specific: Chrompodellids, Evosea, Bryozoa, Echinodermata, and Excavata were absent in the NW region, while Discosea, Amoebozoa, Centroplasthelida, Myxozoa and Tubulinea were not detected in the SW region.

At the ASV level, the SW region exhibited a higher number of unique ASVs than the NW region, with only a few ASVs shared between regions for both prokaryotes and eukaryotes (Fig. 6B). Regarding plastic shape, "fragment" and "line" shared more ASVs between them than with other plastic shapes, likely due to the uneven number of foam and pellet samples compared to the other categories. In terms of plastic polymers, PE and PS harboured more unique prokaryotic ASVs, while PE and PA contained more unique eukaryotic ASVs in both 18S datasets.

## Discussion

This study represents the first comprehensive survey of the marine plastisphere around the Antarctic Peninsula, assessing both species richness and community composition of prokaryotes and eukaryotes. It is also the first study to fully assess the plastisphere of plastics collected from the ocean surface in Antarctica, covering different areas, both near the coast and in the open sea. We show a wide range of taxa living on plastics in Antarctica, with community composition being shaped by regions and not by plastic categories (polymer and shape). This is corroborated by patterns observed for the plastisphere of several other areas, such as the Mediterranean Sea[59], the North and Baltic Seas[19], and the South Pacific[60] and South Atlantic[50] Oceans, in which the geographic region influences the plastisphere composition more than the characteristics of the plastics themselves.

### The Antarctic plastisphere compared with other regions

We detected many benthic species in the Antarctic plastisphere (e.g., the diatoms *Synedra* sp. and *Nitzschia* sp.), which are likely well adapted to a life attached to substrates. As Antarctica does not have any large trees or shrubs, the majority of debris entering marine systems is either man-made or natural debris carried to the region by ocean currents or direct local disposal[2]. The bacterial community composition shown here is similar to what has been observed for the plastisphere of other regions[21] (review of studies from the northern hemisphere), with a dominance of Proteobacteria and Bacteroidetes, specifically Alpha and Gammaproteobacteria[61]. This has also been seen previously in Antarctica, where a microcosm-based colonisation study recently demonstrated the dominance of phyla Pseudomonadota and Bacteroidota, and the class Flavobacteria[32], although the total number of detected phyla in the microcosm was 5, whereas this present study detected 43, suggesting that plastics in natural systems subjected to varied environmental conditions may accumulate a range of bacterial taxa. The majority of prokaryotes and eukaryotes that were observed in this present study on plastics are common taxa from sediments[62], snow[63], and seawater in Antarctica[64]. Here, we have shown that these organisms are currently interacting with this new anthropogenic stressor in the Southern Ocean.

Archaea occurred in low abundance in the Antarctic plastisphere, which was also been shown in the Baltic Sea[23], and areas in the North Pacific[65] and South Atlantic Ocean basins[50,66]. The dominance of Bacteria in the samples evaluated here, as well as those from around the world[21], is likely due to the fact that bacteria are much more abundant than archaea in marine biofilms[67]. Additionally, bacteria are highly interconnected with eukaryotes on plastics[23,68].

The dominance of eukaryotes from the SAR supergroup (Stramenopiles, Alveolata and Rhizaria) and Fungi in the Antarctic plastisphere is aligned with other regions around the globe. For example, when analysing the V4 region of the 18S gene, Kettner et al.[23] found that most of the microeukaryotes living on plastics off the coast of Germany were from the SAR supergroup, Fungi, Metazoa and Chloroplastida. In addition, these groups were also frequent in the plastisphere from the South Pacific[61] and South Atlantic Ocean basins[50,69]. The findings of this current study demonstrate that the Antarctic plastisphere harbours a similar microbial community to other global regions.

### Diversity across Antarctic marine ecosystems

The diversity of microbial marine life in Antarctica is a relatively understudied subject compared to larger organisms, particularly those studies that use multi-marker eDNA metabarcoding[70]. Due to the high rate of endemism, Antarctica is particularly vulnerable to disturbances, and species loss in the Southern Ocean is more likely to be a major contributor to total global biodiversity loss[71]. Therefore, plastics in the ocean, which can serve as substrates for various benthic and planktonic species, could potentially pose a significant threat to these systems, particularly if harbouring disease-causing microorganisms or invasive species. The attachment of diverse taxa to plastic debris can alter natural processes by facilitating the transport of these organisms. As the plastisphere can be a self-sustaining system, allowing many ecological relations to take place[68,72], these miniature ecosystems have been hypothesised to increase the chances of survival of several organism groups if transported between oceanic regions[68].

Many studies have identified dominant prokaryotic phyla like Proteobacteria, Bacteroidetes, and Planctomycetes in Antarctic benthic communities[64], which correspond closely with those found in the plastic samples evaluated here. In addition, they also found various benthic eukaryotic organisms, including Nematoda, Arthropoda and Mollusca groups, which were also observed in the plastisphere samples we evaluated. Moreover, a great diversity of Arthropoda, Bacillariophyta and Annelida was identified in shallow hard-bottom communities from the Western Antarctic Peninsula[70], which are in accordance with the results presented here. This reinforces the role of plastics in hosting benthic species that can now be found floating at the sea surface after attaching to plastics.

Regarding the free-living organisms around the Antarctic Peninsula, Luria et al.[73] observed that bacterial community composition shifts seasonally, with an increased abundance of several taxa associated with phytoplankton blooms during summer, particularly *Polaribacter* species. Indeed, two *Polaribacter* ASVs were among the ten most abundant bacteria we observed within the Antarctic plastisphere. Moreover, Flavobacteriaceae and Rhodobacteraceae are abundant free-living bacterial groups during early summer, but their populations decline in February and March[74]. However, in the plastisphere samples evaluated in this study, which were collected in mid-late February, these families were the most abundant. This raises concerns that plastics may create unique microenvironments, allowing certain species to thrive under conditions that differ from natural aquatic environments. Although research on polar ecosystems is expanding, recent reviews have largely overlooked the impact of plastics and the plastisphere on these ecosystems, highlighting the need for more comprehensive studies in this field.

### Oceanic regions drive community composition

Although some studies have shown differences in the plastisphere community composition according to polymer type, these are mostly based on colonisation experiments. The majority of studies on plastic samples from natural environments with mature biofilms often highlight that the polymer does not seem to influence the diversity and community structure of the plastisphere[69]. This seems to be especially true for mature communities formed on plastics that may have been in the environment for long periods of time[21]. Here, this study evaluated plastics collected from the environment and observed that, indeed, polymer type did not shape the plastisphere in Antarctica. Moreover, recent studies reported no correlation between any polymer and biofouling type on plastics washed up on the shore in the Antarctic Specially protected area Robert Island[74], as well as in mesocosm experiments conducted in Livingston Island[32].

It was previously stated that geographical location, rather than polymer, shapes the plastisphere community structure[59]. In alignment with this statement, this current study observed differences among the NW and SW regions of the Antarctic Peninsula, which could be explained by the local environmental factors such as ocean circulation, sea surface temperature and macronutrients. When describing the macronutrients variability in the Southern Ocean based on a time series spanning from 1996 to 2019, a seasonal macronutrient drawdown was observed for the summer 2017, with higher values for dissolved inorganic nitrogen, phosphate and silicic acid in the Gerlache strait[75]. These authors also showed that the silicic acid/N (Si:N) uptake ratio was higher in the Bransfield strait than in the Gerlache strait. Such results highlight the biogeochemical differences among the two sampled regions of this present study, reinforcing that the environmental factors have an influence on microbial communities[76], including the ones living on plastics.

We observed greater diversity of both prokaryotic and eukaryotic taxa in the SW region. This pattern may be attributed to the relatively warmer waters in the southwest part of the Antarctic Peninsula, which are heavily influenced by the Bellingshausen Sea.

In contrast, the colder and more saline waters in the northern Bransfield Strait are predominantly affected by the Weddell Sea[77] (and also observed in the present study). In the NW region, it was observed that deep-water samples were significantly enriched by Archaea, Plantomycetes and Chloroflexi, whereas shallow-water samples showed a higher contribution of Bacteroidetes, and Proteobacteria (mostly Rhodobacteraceae and Oceanospirillaceae)[76].

We also observed Pseudomonadota and Bacteroidota as the dominant groups in both regions, but archaeal groups and bacteria Chloroflexota, previously found as dominant taxa for deep waters in the NW[76], presented higher abundances on floating plastics from the SW region. The inflow of the Bransfield current in the NW Antarctic Peninsula, along with small-scale vortexes, may shape plastisphere communities. However, strong events such as storms, especially during the winter season, may overcome these barriers and allow plastics—with their associated communities—to travel around the Antarctic continent, as demonstrated in a 7-year plastics dispersal model in Antarctica[2].

### Environmental implications of the plastisphere in Antarctica

Plastics in marine environments, especially in remote regions like Antarctica, serve as vectors for the dispersal of various organisms that might not naturally reach these areas. These communities can disrupt local ecosystems by introducing non-native species[13], altering food webs[78] and impacting biogeochemical cycles[79]. For example, pennate diatoms, which are typically ice-associated, were found adhering to plastic samples, as identified by DNA analysis in this study and by SEM[2]. This indicates a potential for increased dispersal and ecological shifts in areas where it would not naturally thrive.

The Antarctic plastisphere hosts bacterial groups that have been previously described to potentially degrade hydrocarbons and plastics, such as *Tenacibaculum*[80], *Oleispira*[81], *Pseudomonas*[82], *Acidovorax*, *Comamonas* and *Ralstonia*[83,84], *Alcanivorax*[85], *Vibrio*[86] and *Bacillus*[20] species. We also found prokaryotic groups that have been previously reported as potential pathogens, such as species belonging to the genera *Vibrio* and *Acinetobacter*[87], as well as *Staphylococcus*[88]. Apart from harbouring pathogenic species, the plastisphere can also enhance the accumulation of pollutants on plastic debris[89], which exacerbates the ecotoxicological impacts on marine life. Biofilm-covered plastics can mimic the appearance and smell of natural food, leading to increased ingestion of plastics by marine animals[90]. The ingestion of these plastics by marine species could lead to the transfer of harmful pollutants and pathogens, posing a threat to the fragile ecosystems in Antarctica.

Climate change is likely to further exacerbate the ecological impacts of Antarctic microbial communities[91], which can include the plastisphere. Rising temperatures and changes in ocean chemistry could influence the composition and function of plastisphere communities[92,93], as evidenced by studies in other regions[94–96]. In warmer conditions, such as those observed during anomalously warm summers in Antarctica, the biodegradation of plastics may increase. Bacteria species with the potential to degrade plastics, such as *Oleispira* sp., *Alcanivorax* sp., as well as Colwelliaceae and Vibrionaceae species, were indeed found in the present study. Furthermore, the increase in temperature may favour the virulence of pathogens, such as *Vibrio* species[97]. This underscores the urgent need for further research to understand the long-term implications of the plastisphere in Antarctica.

### Conclusion

Using a combination of eDNA-metabarcoding and SEM, we provided a detailed and high-resolution characterisation of the marine plastisphere in the Antarctic Peninsula, revealing a diverse range of prokaryotes and eukaryotes associated with plastics. Given that polar regions typically exhibit lower biodiversity and fewer trophic levels, these findings highlight the role of plastics as a new, mobile, and durable substrate that, in addition to other anthropogenic stressors, have the potential to significantly alter local ecosystems. We demonstrated that many benthic/sessile groups from coastal areas are now living attached to floating plastics in the open ocean in the region. Moreover, several microorganisms within the marine plastisphere possess biotechnological potential, emphasising the need for omics-based approaches to further explore and understand the ecological functions of the Antarctic plastisphere. As plastic pollution increases in the region, we can expect greater availability of artificial substrates for the establishment of diverse species. One of the key recommendations of this study is to include comparative analyses of plastisphere communities between the eastern (Weddell Sea) and the western Antarctic Peninsula, as this is crucial for understanding the differential impacts of climate change on biota in these distinct areas. Key areas for future research also include understanding the limits of species coexistence on plastic surfaces and the impacts of these communities on other species in the Southern Ocean. Addressing these questions will be crucial for predicting and mitigating the long-term ecological consequences of plastic pollution in Antarctica.

### Reporting Summary

Further information on research design is available in the Nature Portfolio Reporting Summary linked to this article.

### Data availability

All DNA sequences used for this research are deposited in the public repository European Nucleotide Archive, accession number PRJEB87624. Additional supporting data, including ASV tables for the 16S, 18S-V4, and 18S-V9 rRNA genes, along with environmental data and detailed information on plastic characteristics for each sampling site, is available on FigShare (https://doi.org/10.6084/m9.figshare.28784807.v1).

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

## Acknowledgements

This work is a contribution of the High Latitude Oceanography Group (GOAL) under the scope of the Brazilian Antarctic Programme (PROANTAR). We thank Carlos Fujita and the crew of the NPo Almirante Maximiano for helping with oceanographic survey and sampling. We thank Dr Thorunn Helgason, Dr Sally James and Dr Peter Aston from the University of York Genomics & Bioinformatics Laboratory, for assistance, support and technical expertise with sequencing. We thank Dr. Lucas Almeida for producing the map in Fig. 1. We thank Mikaele, Lijainah and Nathalia from EQA-FURG for their assistance with FTIR analysis. SEM images were taken at the Electron Microscopy Centre (CEME-Sul, PROPESP-FURG) with the assistance of Rudmar Krumreick and Caroline Ruas. This study was conducted within the activities of project INTERBIOTA, financed by the Conselho Nacional de Desenvolvimento Científico e Tecnológico - CNPq Grant number 407889/2013-2. CNPq also provided research fellowship to ALL (SWE 206250/2017-7), MCP (PQ 312470/2018-5), FK (PQ 435612/2018-2), and ERS (PQ 310597/2018-8) during the development of this study. ALL received a scholarship from the Coordination for the Improvement of Higher Education Personnel (Coordenação de Aperfeiçoamento de Pessoal de Nível Superior – CAPES), which also provided access to the Portal de Periódicos and financial support through Programa de Excelência Acadêmica – PROEX. JDT was funded by a UKRI NERC grant (NE/X012204/1).

## Author contributions

ALL study conception, funding acquisition, sampling, laboratory work and data analysis. M.C.P. studied conception, supervision, funding acquisition and sampling. E.R.S. and C.R.M. funding acquisition and oceanographic survey design. F.K. performed FTIR analysis. J.D.T. funding acquisition, supervision laboratory work and data analysis. ALL wrote the first draft of the paper, and all authors contributed to discussing and editing the manuscript.

## Competing interests

The authors declare no competing interests.
