## [Transparent Peer Review file · Communications Earth & Environment]

Oceanic regions shape the composition of the Antarctic plastisphere

Corresponding Author: Dr Joe Taylor

Version 0:

Decision Letter:

Dear Dr Taylor,

Your manuscript titled "Oceanic provinces shape the composition of the Antarctic plastisphere" has now been seen by 3 reviewers, whose comments are appended below. You will see that they find your work of some potential interest. However, they have raised quite substantial concerns that must be addressed. In light of these comments, we cannot accept the manuscript for publication, but would be interested in considering a revised version that fully addresses these serious concerns.

We hope you will find the reviewers' comments useful as you decide how to proceed.

Should additional work allow you to

- address these criticisms (that is, either to incorporate the suggestions or provide a compelling argument why the point made by the reviewer is not valid, or relevant to the editorial threshold as outlined below)

AND

- meet our editorial thresholds as outlined below,

then we would be happy to look at a substantially revised manuscript.

In the following, we list our main editorial concerns.

- demonstrate how oceanic provinces shape the composition of the Antarctic plastisphere
- fully justify your methodology
- use updated taxonomy for microbial classifications
- ensure up-to-date bioinformatics tools are used where possible

If you choose to take up this option, please either highlight all changes in the manuscript text file, or provide a list of the changes to the manuscript with your responses to the reviewers.

When resubmitting, please provide a point-by-point response to the reviewers' comments. Please submit your responses as a separate file, distinct from your cover letter where you can add responses to the Editors' comments that you do not want to be made available to the reviewers. Word files are preferred. We recommend that any figures, tables or graphs that are included in the response to reviewers are also included in the main article or Supplementary Information.

If the revision process takes significantly longer than three months, we will be happy to reconsider your paper at a later date, as long as nothing similar has been accepted for publication at Communications Earth & Environment or published elsewhere in the meantime.

Please use the following link to submit your revised manuscript, point-by-point response to the reviewers' comments with a list of your changes to the manuscript text (which should be in a separate document to any cover letter), a tracked-changes version of the manuscript (as a PDF file) and any completed checklist:

Link Redacted

Please do not hesitate to contact us if you have any questions or would like to discuss the required revisions further. Thank you for the opportunity to review your work.

Best regards,

Ilka Peeken, PhD
Editorial Board Member
Communications Earth & Environment
orcid.org/0000-0003-1531-1664

Alice Drinkwater
Associate Editor
Communications Earth & Environment
@CommsEarth

EDITORIAL POLICIES AND FORMAT

If you decide to resubmit your paper, please ensure that your manuscript complies with our editorial policies and complete and upload the checklist below as a Related Manuscript file type with the revised article:

Editorial Policy Policy requirements
(Download the link to your computer as a PDF.)

- Behavioural and social science
- Ecological, evolutionary & environmental sciences
- Life sciences

<https://www.nature.com/documents/nr-reporting-summary.zip>

For your information, you can find some guidance regarding format requirements summarized on the following checklist: (<https://www.nature.com/documents/commsj-phys-style-formatting-checklist-article.pdf>) and formatting guide (<https://www.nature.com/documents/commsj-phys-style-formatting-guide-accept.pdf>).

REVIEWER COMMENTS:

Reviewer #1 (Remarks to the Author):

Lacerda and co-workers provide a detailed look at the complex world of the Antarctic plastisphere and reveal how it could impact both biodiversity and the functioning of ecosystems. It is the strength of the current work as we can better grasp how marine plastic pollution affects not only local environments but also wider ecological processes.

It also highlights just how far-reaching the effects of plastic pollution can be, showing how it disrupts food webs, changes habitats, and impacts the resilience of marine life. These insights make it clear that tackling marine plastic pollution is essential to protecting Antarctic marine biodiversity and keeping global marine ecosystems balanced.

The authors claim that they "provided a detailed and high-resolution characterization of the marine plastisphere in Antarctica", which is partially true, as they only investigate the west side of the Antarctic Peninsula, which is one of the most study areas of the Southern Ocean.

Also, claiming the plastic substrates as "trojans" of biodiversity due to their dispersal capacity over the biological communities that can live attached to them is something that has been under investigation for many years (see Fabra et al 2021, <https://doi.org/10.1016/j.scitotenv.2021.149217>).

All that said, this research has the potential to reshape how experts think about both the concepts and technology related to marine plastic pollution. Conceptually, it sheds light on the intricate web of interactions within the Antarctic plastisphere,

challenging the traditional view of pollution as just an isolated problem. Instead, it highlights the deeper connection between plastic debris and marine ecosystems, encouraging a more complete approach to studying and addressing its impacts. On the technological front, this study could pave the way for different approaches and techniques to monitor and analyse microplastic interactions in more extreme environments. This could involve developing more advanced sampling methods, improved molecular tools for identifying the tiny organisms living on and around plastics, or new models that predict how microplastics spread and affect ecosystems. In the long run, this research could lead to more effective strategies for tracking, managing, and reducing plastic pollution worldwide.

However, it seems that the bioinformatics tools, while solid, are a bit outdated, for example the usage of QIIME 1 instead of new tools as QIIME 2 or approaching the biodiversity by using Out instead of ASV, which in this last example is hindering some connections between the sampling points. Also, I wonder if the presence of Archaea found in the samples is artefact related to the bioinformatic analyses of the metabarcoding dataset. The latter is something that authors highlight in the discussion section but without providing any feasible explanation. More information is needed in these questions.

Here some minor comments by line :

Line 190: rarefaction instead of refraction

Line 211: Explain why the authors use Bray-Curtis dissimilarity for prokaryotes and Jaccard distance matrices for eukaryotes instead of unifying the analyses.

I would recommend the unified use of 16/18S rRNA gene instead of only 16/18S throughout the ms. (see the example in the paragraph starting at line 228). Also, I would recommend the author to use the newest taxonomy when refer to prokaryotes – for instance, referring to Pseudomonadota instead of Proteobacteria.

The statements raised in section “Functional potential of prokaryotes in the Antarctic plastisphere” (line 268) are too vague and apparently based in scientific reference, but the authors neither include any of them, not carried out any analyses of the samples in this direction.

The sentence “unique Antarctic taxa were also present, indicating a localised adaptation” deserves a deeper explanation and relation to the approach of the work.

Reviewer #2 (Remarks to the Author):

General Comments:

This study provides a comprehensive molecular survey of microbial diversity on plastics around the Antarctic Peninsula, revealing significant geographic differences in community composition. The findings highlight the complexity of the Antarctic plastisphere and its potential impacts on biodiversity and ecosystem functions. However, the manuscript lacks depth in several key areas, particularly in the description and analysis of the sampling methodology and environmental data.

The manuscript mentions that samples were collected from 12 stations using manta-trawl, but it lacks details on the trawling time, the number of samples collected per transect, and whether all plastics were collected from the same or different transects. This information is crucial for assessing the reliability and significance of the results.

The manuscript should use updated taxonomy for microbial classifications. For instance, Proteobacteria should be referred to as Pseudomonadota, and Bacteroidetes as Bacteroidota. Including a genus plot, Krona plot, or similar data visualization to confirm the results would enhance the manuscript. A Venn diagram showing shared sequences between categories would also be beneficial. The manuscript should specify whether the sequences have been deposited in a public repository, even if they are not yet published.

Besides I suggest to review the writing and avoid talking about “our samples”.

I also suggest the reading of: Monràs-Riera, P; Avila, C; Ballesté E. "Plastisphere in an Antarctic environment: A microcosm approach." *Marine Pollution Bulletin* 208 (2024): 116961.

Specific comments:

Lines 109-113. Environmental parameters such as temperature and salinity, which were collected during sampling, should be included in the manuscript. This data is essential for understanding the context of the microbial communities observed.

Line 126. The authors should provide a detailed account of the plastic sampling results, including the total number of plastics obtained, the distribution of plastics by station, polymer type, plastic shape, and size distribution. This information is necessary to evaluate the significance of the findings.

Line 193-200. The manuscript lacks clarity on the number of samples used for statistical analysis. Indicating the number of plastics for each category and including the sample size (n) for each category (Line 317) would strengthen the statistical analysis.

Line 248. Additionally, "rRNA" should be included after "16S".

Lines 458-459. Instead of stating temperatures as $>0^{\circ}\text{C}$ or $<0^{\circ}\text{C}$, providing a specific temperature range would be more informative. For example, the minimum temperature might be around -2°C , but the maximum above 0°C should be specified.

Reviewer #3 (Remarks to the Author):

The manuscript by Lacerda et al. contributes to our still scarce and fragmentary knowledge on the composition of the marine Antarctic plastisphere (including both prokaryotes and eukaryotes by a eDNA approach), putting the base for a better comprehension of eventual future developments and changes in response to anthropogenic and climate impacts.

In my opinion, the manuscript should be accepted for publication in COMMSENV.

My few comments (mainly addressed to the sampling procedure) are in the annotated manuscript file (attached). Please, check for typesetting errors (some of them are highlighted in yellow). Finally, I think that the manuscript would benefit of a light English revision in some section.

Communications Earth & Environment is committed to improving transparency in authorship. As part of our efforts in this direction, we are now requesting that all authors identified as 'corresponding author' create and link their Open Researcher and Contributor Identifier (ORCID) with their account on the Manuscript Tracking System prior to acceptance. ORCID helps the scientific community achieve unambiguous attribution of all scholarly contributions. You can create and link your ORCID from the home page of the Manuscript Tracking System by clicking on 'Modify my Springer Nature account' and following the instructions in the link below. Please also inform all co-authors that they can add their ORCIDs to their accounts and that they must do so prior to acceptance.

Version 1:

Decision Letter:

Dear Dr Taylor,

Your manuscript titled "Oceanic regions shape the composition of the Antarctic plastisphere" has now been seen by our reviewers, whose comments appear below. In light of their advice we are delighted to say that we are happy, in principle, to publish a suitably revised version in Communications Earth & Environment.

We therefore invite you to revise your paper one last time to address the remaining concerns of our reviewers, which pertain to fully reporting (and acknowledging limitations of) sample numbers, and consistently defining taxonomy. At the same time we ask that you edit your manuscript to comply with our format requirements and to maximise the accessibility and therefore the impact of your work.

EDITORIAL REQUESTS:

*****Please take care to match our formatting and policy requirements. We will check revised manuscript and return manuscripts that do not comply. Such requests will lead to delays. *****

SUBMISSION INFORMATION:

OPEN ACCESS:

Communications Earth & Environment is a fully open access journal. Articles are made freely accessible on publication. For further information about article processing charges, open access funding, and advice and support from Nature Research, please visit <https://www.nature.com/commsenv/open-access>

Link Redacted

Best regards,

Alice Drinkwater, PhD
Associate Editor
Communications Earth & Environment
Consulting Editor
Communications Sustainability

REVIEWERS' COMMENTS:

Reviewer #2 (Remarks to the Author):

This study provides a comprehensive molecular survey of microbial diversity on plastics around the Antarctic Peninsula, revealing significant geographic differences in community composition. The findings highlight the complexity of the Antarctic plastisphere and its potential impacts on biodiversity and ecosystem functions.

In my opinion, the number of samples analyzed per category (e.g., plastic type, region, shape) are insufficient to support strong statistical comparisons since there are many factors which can explain the differences. It is crucial to include the exact number of samples (n) per group in the main text and figure captions—not just in the supplementary material—so readers can evaluate the strength of the statistical inferences. Additionally, sequence counts per sample should be reported. If there is high variation in sequence number between samples, normalization steps should be performed.

The manuscript refers broadly to "bacterial groups" or "taxonomic groups" without consistently defining the taxonomic level (e.g., phylum, class, order). When referring to groups like Pseudomonadota, Synergistota, or Bacteroidota, it should be explicitly stated whether these are phyla, classes, etc. Taxonomic names are inconsistently formatted; italics are not used where appropriate.

The discussion section is disproportionately long compared to the results and would benefit from clearer structure, more focused arguments, and a reduction in speculative statements not based in the data.

Minor Comments:

- Lines 84–96: Text is repeated; please revise for clarity.
- Lines 279–286: Parentheses are inconsistently used or missing.
- Line 296 onwards: Specify the taxonomic level when referring to bacterial groups.
- Line 302: Define "FO" at first mention.
- Line 313 onward: Use italics consistently for all taxonomic names.
- Line 374: Sample size (n) should be included in the main manuscript, especially in any figures or tables representing statistical analysis.

Reviewer #3 (Remarks to the Author):

The authors have addressed my comments and suggestions. In my opinion the manuscript should be accepted for publication.

Review - Point by Point
Manuscript COMMSENV-24-3333

Reviewer #1 (Remarks to the Author):

Lacerda and co-workers provide a detailed look at the complex world of the Antarctic plastisphere and reveal how it could impact both biodiversity and the functioning of ecosystems. It is the strength of the current work as we can better grasp how marine plastic pollution affects not only local environments but also wider ecological processes. It also highlights just how far-reaching the effects of plastic pollution can be, showing how it disrupts food webs, changes habitats, and impacts the resilience of marine life. These insights make it clear that tackling marine plastic pollution is essential to protecting Antarctic marine biodiversity and keeping global marine ecosystems balanced.

Response: We appreciate the careful and helpful reviews provided by Reviewer #1, which greatly helped improve the manuscript.

The authors claim that they “provided a detailed and high-resolution characterization of the marine plastisphere in Antarctica”, which is partially true, as they only investigate the west side of the Antarctic Peninsula, which is one of the most study areas of the Southern Ocean.

Response: We understand that our study does not cover Antarctica entirely, so to clarify this point we replaced Antarctica by “Antarctic Peninsula”. In this sense and in addition to what was proposed by #reviewer1, we have also replaced ‘provinces’ by ‘regions’ in the title, as we understand that oceanic provinces refer to larger dimensions related to biogeochemical features, which is not the case here. We updated this on figures 1, 4, 5 and 6.

Figure 1. Sampling area of floating plastics and their associated plastisphere around the Western Antarctic Peninsula. Sampling sites 1-6: Northwestern region (NW); Sampling sites 7-12: Southwestern region (SW). This map was created with Python software by using the “*matplotlib*” package.

Also, claiming the plastic substrates as “trojans” of biodiversity due to their dispersal capacity over the biological communities that can live attached to them is something that has been under investigation for many years (see Fabra et al 2021, <https://doi.org/10.1016/j.scitotenv.2021.149217>).

Response: We believe that there may have been a misunderstanding here. We are stating that plastics can enhance the dispersal of biological communities over long distances, which we supported with appropriate references. This is distinct from the subject of plastic transport through food webs, as explored in Fabra et al. (2021).

All that said, this research has the potential to reshape how experts think about both the concepts and technology related to marine plastic pollution. Conceptually, it sheds light on the intricate web of interactions within the Antarctic plastisphere, challenging the traditional view of pollution as just an isolated problem. Instead, it highlights the deeper connection between plastic debris and marine ecosystems, encouraging a more complete approach to studying and addressing its impacts.

On the technological front, this study could pave the way for different approaches and techniques to monitor and analyse microplastic interactions in more extreme environments. This could involve developing more advanced sampling methods, improved molecular tools for identifying the tiny organisms living on and around plastics, or new models that predict how microplastics spread and affect ecosystems. In the long run, this research could lead to more effective strategies for tracking, managing, and reducing plastic pollution worldwide. However, it seems that the bioinformatics tools, while solid, are a bit outdated, for example the usage of QIIME 1 instead of new tools as QIIME 2 or approaching the biodiversity by using Out instead of ASV, which in this last example is hindering some connections between the sampling points. Also, I wonder if the presence of Archaea found in the samples is artefact related to the bioinformatic analyses of the metabarcoding dataset. The latter is something that authors highlight in the discussion section but without providing any feasible explanation. More information is needed in these questions.

Response: While we believe our initial pipeline is still valid, and given there are some advantages to using OTUs instead of ASVs for eukaryotes (for example Intragenomic variance in 18S copies), on balance we decided to re-run bioinformatics using DADA2 (Callahan et al. 2016), which is the community standard for most recent microbial ecology studies, and also offers better error correction and denoising of the data.

Results are now shown based on Amplicon Sequence Variants (ASVs). Taxonomic classification has been updated according to the latest SILVA release for prokaryotes (138.2) which takes into account the updated Bacteria taxonomies and the PR² database (5.01) for eukaryotes. Following taxonomic reclassification, we still detected Archaea, comprising 0.5% of the total relative abundance. The presence of Archaea in 16S rRNA metabarcoding datasets is expected when using the 515F/806R primers, as these primers are known to amplify both Bacteria and Archaea, in addition to the fact that the versions of these primers also contain the modifications that enhance their capability to amplify Archaea (Walters et al. 2016). Archaea are present in the marine water column (see reviews Santoro et al. 2019 & DeLong 2021), and have previously been detected in plastisphere communities across the globe (e.g. Debroas et al., 2017; Vaksmaa et al., 2023), including in the Antarctic plastisphere (Monràs-Riera et al., 2024). Our updated classification identified three archaeal groups belonging to the phyla Halobacteriota, Thermoproteota, and Methanobacteriota, which we believe are true members of the marine plastisphere.

1. Walters, W., Hyde, E.R., Berg-Lyons, D., Ackermann, G., Humphrey, G., Parada, A., Gilbert, J.A., Jansson, J.K., Caporaso, J.G., Fuhrman, J.A. and Apprill, A., 2016. Improved bacterial 16S rRNA gene (V4 and V4-5) and fungal internal transcribed spacer marker gene primers for microbial community surveys. *Msystems*, 1(1), pp.10-1128.
2. Santoro, A.E., Richter, R.A. and Dupont, C.L., 2019. Planktonic marine archaea. *Annual review of marine science*, 11(1), pp.131-158.
3. DeLong, E.F., 2021. Exploring marine planktonic archaea: then and now. *Frontiers in Microbiology*, 11, p.616086.
4. Debroas, D., Mone, A., & Ter Halle, A. 2017. Plastics in the North Atlantic garbage patch: A boat-microbe for hitchhikers and plastic degraders. *Science of the Total Environment*, 599, 1222–1232.
5. Vaksmaa, A., Egger, M., Lüke, C., et al. 2022. Microbial communities on plastic particles in surface waters differ from subsurface waters of the North Pacific Subtropical Gyre. *Marine Pollution Bulletin*, 182, 113949.
6. Monràs-Riera, P; Avila, C; Ballesté E. "Plastisphere in an Antarctic environment: A microcosm approach." *Marine Pollution Bulletin* 208 (2024): 116961.

We observed minor changes in the ecological patterns after conducting new bioinformatic analysis, with abundances and frequencies of occurrence changing for some groups, mostly for eukaryotes. The relative abundances of prokaryotes at the phylum level did not change substantially. Methods, results and discussion sections were updated as stated below:

Methods

Line 186-207 - Paragraph changed to: "Primer sequences were removed using Cutadapt (version 1.8) (Martin 2011). Raw paired-end Illumina reads were processed using the DADA2 pipeline (v1.8) in Rstudio to generate amplicon sequence variants (ASVs) from

16S and 18S rRNA gene sequences. Quality filtering and trimming were performed using the filterAndTrim() function, with region-specific parameters to remove low-quality reads and sequencing artefacts. For the 16S, reads were truncated at 220 bp (forward) and 200 bp (reverse), using maxN=0, maxEE=c(2,2), truncQ=2. For 18S-V4, truncation lengths were set to 230 bp (forward) and 200 bp (reverse), with the same filtering thresholds. For 18S-V9, both forward and reverse reads were truncated at 100 bp, with maxEE=c(3,3) to accommodate the shorter amplicon while retaining the same quality parameters. Following filtering, dereplication was performed separately for forward and reverse reads using the derepFastq() function, and error rates were learned using learnErrors(). Reads were then denoised with the dada() function, followed by the merging of paired reads with mergePairs(). Chimeric sequences were removed using the removeBimeraDenovo() function in consensus mode, and a final ASV table was constructed using makeSequenceTable(). Taxonomic assignment was performed using the assignTaxonomy() function with the naïve Bayesian classifier (Wang et al. 2007) and a minimum bootstrap confidence threshold of 80. Prokaryotic (16S) sequences were classified against the SILVA v138.2 database, while eukaryotic (18S rRNA) sequences were classified using the PR² v5.0.1 (Guillou et al. 2013)."

Line 208-212. Additional sentence included: "In the 16S rRNA dataset, eukaryotes, mitochondria, and chloroplasts were excluded before further analysis. Similarly, in both 18S rRNA datasets, unknown domains and certain large metazoans (Salpida and Craniata) were manually removed during analysis, as they were unlikely to be associated directly with the plastics."

Results

Section: DNA Sequence Metrics

Line 272-280 - The number of reads, ASVs, and the mean number of ASVs per sample have been updated.

Section: Prokaryotic Diversity in the Antarctic Plastisphere

Lines 282-298. Number of detected phyla (43 of Bacteria and 3 of Archaea), as well as the (Lines 306-316) names and abundances of the most abundant prokaryotic ASVs were modified according to the updated results.

Figure 3 was replaced and now contains the updated prokaryotic relative abundance and taxonomy.

Figure 3. Relative abundance (RA %) of prokaryotic phyla per plastic sample identified through the 16S rRNA gene according to the region (NW: Northwestern; SW: Southwestern) of the Antarctic Peninsula. 'Others' comprises phyla with lower than 0.005% of relative abundance (Chlamydiota, Aquificota, SAR324 clade (Marine group B), FW113, Fibrobacterota, LCP-89, Sumerlaeota, Zixibacteria, Cloacimonadota, Deferribacterota, Methylospirillum, Thermoplasmata and Caldisepticota).

Section: Eukaryotic Diversity in the Antarctic Plasticsphere

Line 323-369. After the taxonomic reclassification based on the latest PR² release, the taxonomy of eukaryotic groups was updated. The differences in their frequencies of occurrence, according to the molecular marker and location, are now presented in an updated Figure 4. As noted in the previous version of the manuscript, certain eukaryotic groups were exclusively detected by each marker, highlighting the differing resolutions of each and reinforcing our conclusion that multi-marker barcoding is essential for a comprehensive characterization of eukaryotic taxa.

Line 318-362. "The two eukaryotic datasets detected different taxonomic groups. For instance, Gastrotricha, Ichthyophonida, Perkinsea, Platyhelminthes, Radiolaria, Rotifera, Telenemia, unclassified Opisthokonta, and Teleostei were identified only by the 18S-V4, whereas 12 other groups were exclusive to the 18S-V9 dataset: Amoebozoa, Bryozoa, Centroplathelida, Discoba, Euglenozoa, Evosea, Excavata, Haptophyta, Myxozoa, Nebulidia, Picozoa and Porifera, and 'unclassified metazoa'. Twenty-three other eukaryotic groups were identified by both molecular markers: Annelida, Apicomplexa, Arthropoda, Ascidiacea, Cercozoa, Chlorophyta, Choanoflagellata, Chrompodellids, Ciliophora, Cnidaria, Cryptophyta, Ctenophora, Dinoflagellata, Discosea, Echinodermata, Fungi, Gyrista, Mollusca, Nematoda, Rhodophyta, Sagenista, Streptophyta and Tubulinea, along with 'unidentified eukaryotes' (Figure 4; See supplementary material for

details). In both 18S datasets, only seven groups of eukaryotes presented FO higher than 50%. Arthropoda (primarily crustaceans), Gyrista (mainly diatoms), Fungi, Dinoflagellata, Ciliophora, and Streptophyta were consistently present (FOs between 52% and 96%) in both the 18S-V4 and 18S-V9 datasets, although their frequencies varied depending on the marker. For example, Arthropoda was the only group with FO 100% in the 18S-V9 dataset, while Gyrista was the only group with FO 100% in the 18S-V4 dataset. In addition to these groups, Chlorophyta was more abundant on the 18S-V4 (FO 61%), while Apicomplexa (FO 81%) was prevalent in samples identified with the 18S-V9 rRNA. Many diatom ASVs matched species previously described in the NCBI for the Southern Ocean or polar regions, such as the benthic diatom *Navicula glaciei* (Genbank access number EF106789.1), *Chaetoceros socialis* (KX253957.1), *Corethron inerme* (AJ535180.1) and *Porosira glacialis* (ON888453.1). The 18S-V9 marker dataset showed many “unidentified/uncultured” eukaryotes (FO 81%), whereas this group was only observed in 39% of samples within the 18S-V4 dataset. When considering the most frequent eukaryotic ASVs, the dinoflagellate *Cladocopium* sp. (formerly known as *Symbiodinium* sp.), the crustacean *Carpilius* sp., as well as an unidentified Ctenophora and the diatom *Grammonema* sp. (formerly known as *Fragilaria*) stood out in the 18S-V4 dataset. Meanwhile, for the 18S-V9 dataset, fungi species from genera *Aspergillus* and *Sterigmatomyces*, and the pennate diatom *Synedra* (*Fragilariaceae* family) were the most frequent ASVs. We also identified microeukaryotes described as animal parasites, such as the parasitic alveolate *Cephaloidophora* sp., and ciliate species belonging to the *Epistylis* and *Vorticellides* genera. Potential harmful fungi associated with plastics in Antarctica are not reported here, since they were described in detail in Lacerda et al. (2020).”

Frequency of occurrence of eukaryotes (FO%)

Figure 4. Percentage of eukaryotes frequency of occurrence (FO) of eukaryotic groups from the marine Antarctic plastisphere, identified through the 18S rRNA gene (V4 and V9 regions), separated by region (Northwestern and Southwestern). For better visualisation, only groups with FO higher than 10% are shown for the 18S-V4 and 18S-V9 in the Southwestern, and 18S-V9 Northwestern regions.

Section “Species richness and community according to locations and plastic categories”

Overall, species richness did not significantly vary according to location for either prokaryotes or eukaryotes (as previously stated). Moreover, the community structure followed the same pattern shown in the previous submission, i.e variations on community structure only according to regions for prokaryotes and eukaryotes, with no variations according to plastic shape or polymers. All pseudo-F and p-values were updated

accordingly (Line 365-369, and line 382-387). Figure 5 was updated with the number of ASVs per categories.

Figure 5. Mean number of observed ASVs per plastic sample from the Antarctic Peninsula, according to geographic region (Northwestern and Southwestern), polymer type (PA -polyamide; PE - polyethylene; PP - polypropylene; PS - polystyrene; PU - polyurethane) and plastic shape (foam, fragment, line and pellet).

In this updated version of the manuscript, we have also provided more details on the variations in groups occurrence between regions for both prokaryotes and eukaryotes, as shown in the following paragraphs:

Newly added - Line 388-400. “For eukaryotes identified using the 18S-V4 marker, fewer taxonomic groups were detected in the NW region compared to the SW region. Specifically, Nematoda, Chrompodellids, unidentified Opisthokonta, Gastrotricha, Radiolaria, Discosea, Perkinsea, Cryptophyta, Ichthyophonida, Apicomplexa, Annelida, Echinodermata, and Platyhelminthes were absent from the NW region. In contrast, the SW region contained all eukaryotic groups identified with this marker, except for Ascidiacea. Among the eukaryotic groups detected using the 18S-V9 marker, a total of 31 groups were commonly observed in both the NW and SW regions, although their frequencies of occurrence varied by region. However, some groups were region-specific: Chrompodellids, Evosea, Bryozoa, Echinodermata, and Excavata were absent in the NW region, while Discosea, Amoebozoa, Centroplasthelida, Myxozoa, and Tubulinea were not detected in the SW region.

Results:

Here some minor comments by line:

Line 190: rarefaction instead of refraction

Response: Corrected accordingly (Line 216).

Line 211: Explain why the authors use Bray-Curtis dissimilarity for prokaryotes and Jaccard distance matrices for eukaryotes instead of unifying the analyses.

Response: We appreciate the point raised here. The Bray-curtis distance is abundance weighted, whereas Jaccard matrices are not. Given the high variability in 18S rRNA gene copy number across eukaryotic taxa, and the prevalence of multicellular organisms in the dataset, using abundance-weighted measures in metabarcoding analyses can be problematic. For multicellular taxa, sequence read counts do not reliably reflect biomass or the number of individuals, potentially making abundance-based statistics misleading when interpreting Eukaryotic community data. We now give a brief explanation for this in the manuscript, and include the appropriate reference:

Line 226-230. “For eukaryotes, we used unweighted Jaccard matrices instead of Bray-Curtis distances to focus on presence rather than abundance. This approach accounts for the considerable variation in 18S rRNA gene copy numbers among different eukaryotic species, as well as differences in relative abundance between single-cellular and multicellular organisms (Gong & Marchetti 2019)”.

1. Gong, W., & Marchetti, A. (2019). Estimation of 18S Gene Copy Number in Marine Eukaryotic Plankton Using a Next-Generation Sequencing Approach. *Frontiers in Marine Science*, 6, 219. <https://doi.org/10.3389/fmars.2019.00219>

I would recommend the unified use of 16/18S rRNA gene instead of only 16/18S throughout the ms. (see the example in the paragraph starting at line 228).

Response: Corrected accordingly throughout the manuscript.

Also, I would recommend the author to use the newest taxonomy when refer to prokaryotes –for instance, referring to Pseudomonadota instead of Proteobacteria.

Response: We have reclassified prokaryotic sequences with the Silva version 138.2, and the eukaryotic sequences with the PR² version 5.0.1, which has updated taxonomy.

Added reference for the PR²:

1. Guillou, L., Bachar, D., Audic, S., Bass, D., Berney, C., Bittner, L., Boute, C. et al. 2013. The Protist Ribosomal Reference database (PR2): a catalog of unicellular eukaryote Small Sub-Unit rRNA sequences with curated taxonomy. *Nucleic Acids Res.* 41:D597–604.

The statements raised in section “Functional potential of prokaryotes in the Antarctic plastisphere” (line 268) are too vague and apparently based in scientific reference, but the authors neither include any of them, nor carried out any analyses of the samples in this direction.

Response: Our goal was to report microorganisms that were previously described as either hydrocarbons/plastic degraders or pathogens. Considering we have retrieved this information from the literature in a non-systematic way, and as our data does not directly support functional analysis, we moved such statements to the discussion section, including the appropriate references as suggested by reviewer 1 and reviewer 3. The paragraph has now been revised to read as follows:

Line 583-591. “The Antarctic plastisphere hosts bacterial groups that have been previously described to potentially degrade hydrocarbons and plastics, such as *Tenacibaculum* spp. (Sekiguchi et al., 2011), *Oleispira* sp. (Urbanek et al. , 2018), *Pseudomonas* spp. (Balasubramanian et al., 2010), *Acidovorax* sp., *Comamonas* sp., *Ralstonia* sp. (Biki et al., 2021,; Ryan et al., 2007), *Alcanivorax* sp., (Delacuvellerie et al., 2019), *Vibrio* spp. (Raghul et al., 2014), and *Bacillus* spp. (Wright et al., 2020b). We also found prokaryotic groups that have been previously reported as potential pathogens, such as species belonging to the genera *Vibrio* and *Acinetobacter* (Kirstein et al., 2016), as well as *Staphylococcus* (Liu, 2009).”

The following references were included:

1. Sekiguchi T, A Saika, K Nomura, T Watanabe, T Watanabe, Y Fujimoto, M Enoki, T Sato, C Kato, H Kanehiro. 2011. Biodegradation of Aliphatic Polyesters Soaked in Deep Seawaters and Isolation of Poly(ϵ -Caprolactone)-Degrading Bacteria. *Polymer Degradation and Stability* 96 (7): 1397–1403. doi:10.1016/j.polymdegradstab.2011.03.004.
2. Urbanek AK., W Rymowicz, AM Mirończuk. 2018. Degradation of Plastics and Plastic-Degrading Bacteria in Cold Marine Habitats. *Applied Microbiology and Biotechnology* 102 (18): 7669–78. doi:10.1007/s00253-018-9195-y.
3. Biki, S.P., S Mahmud, S Akhter, MJ Rahman, JJ Rix, MAA Bachchu, M Ahmed. 2021. Polyethylene Degradation by *Ralstonia* sp. Strain SKM2 and *Bacillus* sp. Strain SM1 Isolated from Landfill Soil Site. *Environmental Technology & Innovation* 22: 101495. doi:10.1016/j.eti.2021.101495.
4. Delacuvellerie A, V Cyriaque, S Gobert, S Benali, R Wattiez. 2019. The Plastisphere in Marine Ecosystem Hosts Potential Specific Microbial Degraders Including *Alcanivorax Borkumensis* as a Key Player for the Low-Density Polyethylene Degradation. *Journal of Hazardous Materials*, 120899. doi:10.1016/j.jhazmat.2019.120899.
5. Raghul SS, SG Bhat, M Chandrasekaran, V Francis, ET Thachil. 2014. Biodegradation of Polyvinyl Alcohol-Low Linear Density Polyethylene-Blended Plastic Film by Consortium of Marine Benthic Vibrios. *International Journal of Environmental Science and Technology* 11 (7): 1827–34. doi:10.1007/s13762-013-0335-8.
6. Wright RJ, MGI Langille, TR Walker. 2020. Food or Just a Free Ride? A Meta-Analysis Reveals the Global Diversity of the Plastisphere. *The ISME Journal*: 1–18. doi:10.1038/s41396-020-00814-9.
7. Kirstein IV, S Kirmizi, A Wichels, A Garin-fernandez, R Erler, L Martin. 2016. Dangerous Hitchhikers? Evidence for Potentially Pathogenic *Vibrio* spp. on Microplastic Particles. *Marine Environmental Research* 120, 1–8. doi:10.1016/j.marenvres.2016.07.004.
8. Liu, G. Molecular Pathogenesis of *Staphylococcus aureus* Infection. *Pediatr. Res.*, 65, 71–77 (2009). <https://doi.org/10.1203/PDR.0b013e31819dc44d>.

The sentence “unique Antarctic taxa were also present, indicating a localised adaptation” deserves a deeper explanation and relation to the approach of the work.

Response: This sentence was part of an outdated version of the manuscript, in which we discussed differences between the plastisphere of Antarctica and that of our paper in the Atlantic Ocean (Lacerda, et al. 2022 *Sci. Total Environ*, 805, p.150186). However,

considering the limitations of analysing metabarcoding data at the species level, we chose to classify the data at a higher taxonomic level in this updated version of the manuscript. Therefore, we removed the aforementioned sentence from the manuscript, as it does not support the data presented.

#####

Reviewer #2 (Remarks to the Author):

General Comments:

This study provides a comprehensive molecular survey of microbial diversity on plastics around the Antarctic Peninsula, revealing significant geographic differences in community composition. The findings highlight the complexity of the Antarctic plastisphere and its potential impacts on biodiversity and ecosystem functions. However, the manuscript lacks depth in several key areas, particularly in the description and analysis of the sampling methodology and environmental data.

Response: We have addressed all the points raised by the reviewer#2, including a more detailed description of the sampling methodology and environmental data as detailed in the following responses.

The manuscript mentions that samples were collected from 12 stations using manta-trawl, but it lacks details on the trawling time, the number of samples collected per transect, and whether all plastics were collected from the same or different transects. This information is crucial for assessing the reliability and significance of the results.

Response: Line 96-100. We have now included details on the trawling duration. “Samples were collected from 12 stations between 61° and 64°S using a manta net with a 100 cm × 21 cm mouth and a 330 µm mesh. At each station, the net was deployed from the windward side of the vessel via a large A-frame and trawled at a speed of 2.5–3.5 knots for 15–55 minutes, depending on weather and logistical conditions.”

Line 273-278. Additionally, we clarified the number of samples collected per transect in the 'DNA sequence metrics' section, specifying the number of successful samples per molecular marker within each geographic region (NW and SW). We also included detailed information per sampling site in the Supplementary Material (Supplementary material, including ASV tables for the 16S, 18S-V4, and 18S-V9 rRNA genes, along with environmental data and detailed information on plastic characteristics for each sampling site, is available on FigShare

https://figshare.com/articles/dataset/Supplementary_Material_Lacerda_et_al_The_Antarctic_Plastisphere/28784807?file=53626958).

The manuscript should use updated taxonomy for microbial classifications. For instance, Proteobacteria should be referred to as Pseudomonadota, and Bacteroidetes as Bacteroidota. Including a genus plot, Krona plot, or similar data visualization to confirm the results would enhance the manuscript. A Venn diagram showing shared sequences between categories would also be beneficial. The manuscript should specify whether the sequences have been deposited in a public repository, even if they are not yet published.

Response: As stated in the previous comments to Reviewer #1, taxonomic classification was updated. In addition, we accepted the suggestion made by Reviewer#2 and added new venn diagrams at ASV level per category (region, polymer type and shape) (updated Figure 6 below). However, while we appreciate phyla level figures may be quite broad, genus level figures are likely to be too complex to display in a readable and interpretable way; therefore, we provide the full ASV tables as supplementary information with the detailed taxonomy and DNA sequences. We discuss specific genera of interest in both the results and discussion. We have included the following results referring to the Venn diagrams:

Line 401-408. “At the ASV level, the SW region exhibited a higher number of unique ASVs than the NW region, with only a few ASVs shared between regions for both prokaryotes and eukaryotes (Figure 6). Regarding plastic shape, ‘fragment’ and ‘line’ shared more ASVs between them than with other plastic shapes, likely due to the uneven number of foam and pellet samples compared to the other categories. In terms of plastic polymers, PE and PS harboured more unique prokaryotic ASVs, while PE and PA contained more unique eukaryotic ASVs in both 18S datasets.”

Additionally, we added the section “Data availability” to state that sequences are now deposited in the public repository European Nucleotide Archive accession number: PRJEB87624.

A) Community structure by region

B) Shared and unique ASVs by category

Figure 6. A) Non-metric multidimensional scaling (NMDs) plots of the marine plastisphere from the Antarctic Peninsula based on the Bray-Curtis distance matrix for the rRNA 16S, and on the Jaccard distance matrix for the 18S-V4 and 18S-V9 rRNA datasets, according to region (NW: Northwestern and SW: Southwestern).

B) Shared and unique ASVs between regions (NW and SW), polymer type (PA - polyamide; PE - polyethylene; PP - polypropylene; PS - polystyrene; PU - polyurethane) and plastic shape (foam, fragment, line and pellet).

Besides I suggest to review the writing and avoid talking about “our samples”.

Response: We have accepted the suggestion and replaced “our samples” by “the collected samples” or “samples evaluated in the present study”.

I also suggest the reading of: Monràs-Riera, P; Avila, C; Ballesté E. "Plastisphere in an Antarctic environment: A microcosm approach." *Marine Pollution Bulletin* 208 (2024): 116961.

Response: Thank you for suggesting the work by Monràs-Riera et al. (2024). We likely missed this paper because it had not yet been published at the time of our manuscript submission. After carefully reviewing it, we have incorporated a discussion of its findings in comparison with our own results. The updated information is as follows:

Line 67-71 (Introduction). “Two further recent studies have looked at the colonisation of plastics by microorganisms in the Ross Sea (Caroppo et al. 2022) and in microcosm experiments on Livingston Island, in the South Shetlands archipelago (Monràs-Rieira et al. 2024), both using molecular techniques”.

Line 441-447 (Discussion). “This has also been seen previously in Antarctica, where a microcosm-based colonisation study recently demonstrated the dominance of phyla Pseudomonadota and Bacteroidota, and the class Flavobacteria (Monràs-Rieira et al. 2024), although the total number of detected phyla in the microcosm was 5, whereas this present study detected 43, suggesting that plastics in natural systems subjected to varied environmental conditions may accumulate a range of Bacterial taxa.”

Line 530-534 (Discussion). Moreover, recent studies reported no correlation between any polymer and biofouling type on plastics washed up on the shore in the Antarctic Specially Protected Area Robert Island (Johnston et al. 2024), as well as in mesocosm experiments conducted in Livingston Island (Monràs-Rieira et al. 2024).

Specific comments:

Lines 109-113. Environmental parameters such as temperature and salinity, which were collected during sampling, should be included in the manuscript. This data is essential for understanding the context of the microbial communities observed.

Response: We have included the required information. Line 103-109 (Methods) “Salinity and sea temperature were recorded concurrently with plastic sampling at each site. HPLC-derived measurements of chlorophyll-a concentrations were obtained from Ferreira et al. (2024), following the methodology established by the Brazilian High-Latitude Oceanography Group (GOAL-FURG). Detailed information on the environmental parameters for each sampling site is provided in the Supplementary Material”

Line 249-257 (Results). “Environmental Characterization of the Studied Regions – Warmer surface waters were observed in the Gerlache Strait (SW region), with temperatures ranging from 1.56°C to 3.23°C. In contrast, the Bransfield Strait (NW region) exhibited colder surface waters, with temperatures ranging from 1.03°C to 2.48°C (see Supplementary Material). Regarding salinity, the NW region displayed a range from 33.9 to 34.2, while the SW region showed a variation ranging from 33.5 to 33.9. In terms of chlorophyll-a concentrations, the NW region had lower values, ranging from 0.19 to 1.14 mg.m⁻³ (mean 0.52, SE ± 0.17), whereas the SW region exhibited higher concentrations, ranging from 0.26 to 4.65 mg.m⁻³ (mean 2.10, SE ± 0.71).”

Line 126. The authors should provide a detailed account of the plastic sampling results, including the total number of plastics obtained, the distribution of plastics by station, polymer type, plastic shape, and size distribution. This information is necessary to evaluate the significance of the findings.

Response: General information on the qualitative and quantitative characterization of plastics is published in Lacerda et al., 2019 (as described in the manuscript), and therefore was not included in the present study on the microbial communities inhabiting these plastics. However, we have now provided detailed information in the Supplementary Material about the plastics used for this study.

Line 193-200. The manuscript lacks clarity on the number of samples used for statistical analysis. Indicating the number of plastics for each category and including the sample size (n) for each category (Line 317) would strengthen the statistical analysis.

Response: As stated in the previous response, this information is now available in the Supplementary Material. The sample size for each category varied depending on the molecular marker dataset, as the number of successfully amplified samples differed. We have included the sample size range for each category and region in the supplementary material. For shape category: Fragment 7-13; Line 5-9; Foam and Pellet 2 each. For polymer category: PE 3-4; PA: 3; PU: 2-3; PS: 2. There was only one polypropylene

sample, therefore it was excluded from the statistical analysis based on polymer type as stated in the manuscript.

Line 248. Additionally, "rRNA" should be included after "16S".

Response: Modified accordingly throughout the manuscript.

Lines 458-459. Instead of stating temperatures as $>0^{\circ}\text{C}$ or $<0^{\circ}\text{C}$, providing a specific temperature range would be more informative. For example, the minimum temperature might be around -2°C , but the maximum above 0°C should be specified.

Response: Line 247-257. We have created a new section "*Environmental characterization of the studied regions*" and provided details on temperature, salinity and chlorophyll-a ranges. In addition, measurements per sampling site are now provided on supplementary material.

#####

Reviewer #3 (Remarks to the Author):

The manuscript by Lacerda et al. contributes to our still scarce and fragmentary knowledge on the composition of the marine Antarctic plastisphere (including both prokaryotes and eukaryotes by a eDNA approach), putting the base for a better comprehension of eventual future developments and changes in response to anthropogenic and climate impacts.

In my opinion, the manuscript should be accepted for publication in COMMSENV.

My few comments (mainly addressed to the sampling procedure) are in the annotated manuscript file (attached). Please, check for typesetting errors (some of them are highlighted in yellow). Finally, I think that the manuscript would benefit of a light English revision in some section.

Response: Thank you very much for the valuable reviews and suggestions to improve our manuscript. Please see below our reply to your comments. The highlighted typos and misspelled words have been corrected throughout the manuscript. Detailed responses to address specific procedures are provided in the following responses.

Introduction- Plastics have been also detected in biota, e.g. marine sponges

Response: Line 33: "(...) and biota (Ivar do Sul et al. 2011)"

Please, update this section (introduction about the plastisphere studies conducted in Antarctica) with recent papers on Antarctic marine plastisphere. The same applies to discussion

Response: Both introduction and discussion were updated as detailed in the previous responses.

Please, add information about the hauling speed, duration of the sampling, and the distance covered during sampling, and so on.

Response: We have now added the following paragraph as also stated in a previous response to reviewer#1.

Line 96-100: “Samples were collected from 12 stations between 61° and 64°S using a manta net with a 100 cm × 21 cm mouth and a 330 µm mesh. At each station, the net was deployed from the windward side of the vessel via a large A-frame and trawled at a speed of 2.5–3.5 knots for 15–55 minutes, depending on weather and logistical conditions”

Sterile? aluminium bags

Response: Aluminium bags were not sterile, but plastics were rinsed in sterile artificial seawater before DNA extraction to remove loosely associated organisms, i.e. organisms that were not attached to the biofilm on plastics. In addition, all of the genera we identified consisted of marine species, further confirming that we described plastic-associated marine organisms.

How did you separate plastic pieces when frozen all together?

Response: After thawing the samples in sterile artificial seawater, we used sterile tweezers to carefully pick up each floating plastic piece and rinsed it in clean artificial seawater to remove loosely associated organisms before transferring it to the extraction tubes.

Biomass estimation? It is not clear to me.

Response: To avoid confusion, we have replaced "biomass" by organic matter in the updated manuscript. What we meant by “(...) for manual separation of floating plastic pieces and biomass” is that we placed the samples separately in a sterile container filled with artificial sterile saltwater and manually separated the floating plastics from the zooplankton and algae that were collected alongside the plastics during trawling, following the methods applied by Reisser et al. (2014).

Did you collect a total of 32 pieces of plastics? If yes, this info should be moved to results. The same for FTIR samples.

Response: No, 32 samples were used for this description of microbial communities attached to plastics in Antarctica. This is part of a larger study developed by our research team (GOAL), and the qualitative and quantitative characterization of plastics are published in Lacerda et al. (2019) as we stated in the manuscript. In addition, 28 out of the 32 samples were submitted to FTIR analysis to verify their polymeric composition. These are not results from this manuscript, and therefore we believe that they are more appropriately placed in the Methods section.

(Results - SEM section) Was it applied to the 14 additional samples?

Response: Yes, all samples showed many groups of organisms living within the plastisphere. We highlighted in the updated manuscript that 'additional 14 samples' were submitted to SEM analysis.

21 (out of ?)

Response: 21 out of 32 samples as we stated on methods (line 140).

In my opinion, as the authors did not carry a functional metabolic profiling, this text (being a comment) should be moved in the discussion.

Response: After agreeing with the two reviewers who highlighted this point, we moved the sentence to the Discussion section and added the relevant references, as detailed in our response above.

The authors should consider to cite "province", Polymer and shape only one time per graph typology

Response: Changed accordingly.

Review - Point by Point 2 ROUND
Manuscript COMMSENV-24-3333

Please note that line numbers in responses correspond to the updated manuscript.

Editor: We therefore invite you to revise your paper one last time to address the remaining concerns of our reviewers, which pertain to fully reporting (and acknowledging limitations of) sample numbers, and consistently defining taxonomy. At the same time we ask that you edit your manuscript to comply with our format requirements and to maximise the accessibility and therefore the impact of your work. Please take care to match our formatting and policy requirements. Please outline your response to each request in the right hand column. Please upload the completed table with your manuscript files as a Related Manuscript file.

Response: We appreciate the opportunity to revise our manuscript. As requested by Reviewer #2, we have updated the supplementary information to include details on the number of samples per category in the caption of Figure 5. We had already acknowledged the limitations related to sample size in our discussion of the results. For instance, in lines 336-337, we stated: “(...) likely due to the uneven number of foam and pellet samples compared to the other categories”

The Manuscript was updated to match the journal’s formatting and policy requirements.

REVIEWERS' COMMENTS:

Reviewer #2

In my opinion, the number of samples analyzed per category (e.g., plastic type, region, shape) are insufficient to support strong statistical comparisons since there are many factors which can explain the differences. It is crucial to include the exact number of samples (n) per group in the main text and figure captions—not just in the supplementary material—so readers can evaluate the strength of the statistical inferences. Additionally, sequence counts per sample should be reported. If there is high variation in sequence number between samples, normalization steps should be performed.

Response: As requested, we have now included the sample numbers for each category in the figure legends. These values were already mentioned in the main text under the section ‘DNA sequence metrics’ for the corresponding regions. Additionally, we have added the following sentence on data normalization in lines 183-184: “To assess beta diversity, we normalized ASV read counts by converting them into relative abundance values.” Furthermore, we have included details about sequence counts per sample in the main text (Lines 226–235), according to each marker, as follows: “For the 16S dataset, the number of reads ranged from 885 to 55,923. The number of ASVs per sample ranged from 10 to 147 and from 9 to 109 in the 18S-V4 and 18S-V9 datasets, respectively. The number of reads ranged from 479 to 63,520 for 18S-V4, and from 1,156 to 89,941 for 18S-V9.”

The manuscript refers broadly to "bacterial groups" or "taxonomic groups" without consistently defining the taxonomic level (e.g., phylum, class, order). When referring to groups like Pseudomonadota, Synergistota, or Bacteroidota, it should be explicitly stated whether these are phyla, classes, etc. Taxonomic names are inconsistently formatted; italics are not used where appropriate.

Response: For Bacteria, we consistently stated the taxonomic rank before mentioning the corresponding taxa throughout the manuscript. We have carefully reviewed the text to ensure this convention is applied uniformly, with two noted exceptions: in line 240, we added “the class” before Gammaproteobacteria for clarity; and in line 311, we replaced “taxa” with “phyla” for accuracy.

Regarding Eukaryotes, to address taxonomic inconsistencies across diverse groups, we have opted to use the broader term “Eukaryotic groups” rather than referring to a specific taxonomic level. This terminology is widely used in the literature when dealing with heterogeneous eukaryotic lineages (e.g., Oldenburg et al., 2024; Eckmann et al., 2024). Additionally, we have confirmed that all genus and species names are correctly italicized, while names of phyla and other higher taxonomic ranks are not, in accordance with guidelines from international taxonomic authorities (which is also followed in the following listed references)

References:

1. Oldenburg et al. (2024). *Commun. Earth Environ.*, 5, 266. <https://doi.org/10.1038/s43247-024-01782-0>
2. Eckmann et al. (2024). *Commun. Earth Environ.*, 5, 266. <https://doi.org/10.1038/s43247-024-01395-7>

The discussion section is disproportionately long compared to the results and would benefit from clearer structure, more focused arguments, and a reduction in speculative statements not based in the data.

Response: Regarding the suggestion for a clearer structure in the Discussion, we would like to emphasize that the section is already organized into four clearly defined parts. We believe this format provides a logical and transparent flow to the arguments presented. The structure is as follows:

Section 1 – The Antarctic plastisphere compared with other regions:

Comparison of our data with global plastisphere datasets.

Section 2 – Diversity across Antarctic marine ecosystems:

Comparison between natural communities and those associated with plastic surfaces.

Section 3 – Oceanic regions drive community composition:

Discussion of the manuscript’s main findings on biogeographic patterns.

Section 4 – Environmental implications of the plastisphere in Antarctica:

Ecological interpretation of the plastisphere in the studied region.

Regarding the request for more focused arguments and reduction of speculative statements, we find this comment somewhat vague and difficult to address directly without further clarification. Nevertheless, we have critically revised the discussion to improve conciseness and remove statements that could be perceived as overly speculative. Specifically, we removed the following sentences:

“These recent works also point out significant differences in microbial communities across various locations along the Western Antarctic Peninsula, suggesting high levels of endemism and variability in community composition (Angulo-Preckler et al. 2023, Fonseca et al. 2022). However, the transport of plastics across different regions threatens these natural boundaries, as floating plastics can introduce non-native species, which could lead to further ecological imbalances.”

“The marine plastisphere in Antarctica presents significant ecological implications due to its ability to alter natural ecological interactions.”

“such as those observed during anomalously warm summers in Antarctica (...), potentially releasing even more pollutants and altering microbial communities”

Minor Comments:

- Lines 84–96: Text is repeated; please revise for clarity.

Response: Line 86, we removed the phrase “the Brazilian High-Latitude Oceanography Group” as it had already been mentioned earlier. Additionally, in line 66, we deleted the phrase “the prokaryotic and eukaryotic”, and in line 67, we replaced “full diversity of life inhabiting plastics” by “plastisphere” to improve the flow and conciseness of the text.

- Lines 279–286: Parentheses are inconsistently used or missing.

Response: We did not identify any missing parentheses in the specified lines. However, to enhance clarity and improve the flow of the text, we added “Figure 2” before the corresponding letters within the parentheses (line 219). Additionally, we corrected the use of parentheses in lines 139, 141, 143.

- Line 296 onwards: Specify the taxonomic level when referring to bacterial groups.

Response: Stated in a previous response.

- Line 302: Define "FO" at first mention.

Response: There seems to be a misunderstanding, as we did include a definition of FO - Frequency of Occurrence - at its first mention in the manuscript (line 249-250).

- Line 313 onward: Use italics consistently for all taxonomic names.

Response: We have used italics for genera and species throughout the manuscript.

- Line 374: Sample size (n) should be included in the main manuscript, especially in any figures or tables representing statistical analysis.

Response: Updated Figure 5 caption now contains number of samples by category “Mean number of observed ASVs per plastic sample from the Antarctic Peninsula, according to geographic region (Northwestern n = 14, and Southwestern n = 12), polymer type (PA – polyamide n = 3; PE – polyethylene n = 4; PP – polypropylene n = 1*; PS – polystyrene n = 2; PU – polyurethane n = 3) and plastic shape (foam n = 2; fragment n = 13; line n = 9, and pellet n = 2). *Not considered for statistical analyses.”

Additional modifications made by authors:

Line 168: We added the word “database” after “PR2 v5.0.1” for clarity.

Line 358: The term “Bacteria” was replaced with “the bacterial” for grammatical consistency.

All figure captions have been grouped at the end of the manuscript file, and a title has been added to each figure as requested.

The author affiliations section was revised to comply with the journal’s formatting requirements.

Keywords were removed, as the journal does not include them in published articles.

The reference list was revised and corrected in accordance with the journal’s citation style.

Terminology update: The phrase “supplementary material” was replaced with “supplementary data” throughout the manuscript, as per the journal’s preferred terminology.

Oceanic provinces shape the composition of the Antarctic plastisphere

Ana L. Lacerda^{1,2*}, Maíra C. Proietti^{1,3}, Felipe Kessler⁴, Carlos R. Mendes¹,
Eduardo R. Secchi¹, Joe D. Taylor^{5*}

¹Instituto de Oceanografia, Universidade Federal do Rio Grande (FURG), Rio Grande, Brazil; ²Association Expédition MED, Questembert, France; ³The Ocean Cleanup, The Netherlands; ⁴Escola de Química e Alimentos, Universidade Federal do Rio Grande (FURG), Rio Grande, Brazil; ⁵UK Centre for Ecology and Hydrology, Wallingford, UK.

*corresponding authors: ana.luzia-lacerda@imev-mer.fr; joetay@ceh.ac.uk

Abstract

Antarctica, once considered pristine, is increasingly threatened by plastic pollution, with debris found in its waters, sediments, and sea ice. Here we provide a comprehensive molecular survey of both prokaryotic and eukaryotic diversity on plastics around the Antarctic Peninsula, addressing a gap in existing research. Using eDNA metabarcoding, we identified diverse communities, with Proteobacteria and Bacteroidetes dominating prokaryotes, while diatoms, dinoflagellates, and fungi were prevalent among eukaryotes. Geographic location significantly influenced community composition, with differences between the northwestern and southwestern regions of the Antarctic Peninsula. Polymer type and plastic shape did not significantly impact species richness or community structure. These findings offer new insights into the complexity of the Antarctic plastisphere, highlighting potential impacts on biodiversity, ecosystem functions, and the broader implications of marine plastic pollution.

Keywords: Plastics, Microplastics, Biofilm, Southern Ocean, eDNA

Introduction

Antarctica was once considered a pristine environment, but several studies have now highlighted that this region contains a range of pollutants, including plastic pollution in surface waters (Suaria et al. 2020; Lacerda et al. 2019; Waller et al. 2017), water column (Isobe et al., 2017), deep-sea sediments (Cunningham et al. 2020; van-Cauwenberghe et al. 2013) and sea ice (Kelly et al. 2020). A thirty-year monitoring study of debris in the Southern Ocean identified variable trends in debris concentration over time, and plastics represented more

Commented [u1]: Plastics have been also detected in biota, e.g. marine sponges

than 80% of debris items in two locations of the Scotia Sea (Waluda et al. 2020).
The impacts of plastic on Antarctic wildlife have been increasingly reported, and
include ingestion by a range of species, from small benthic animals to megafauna
(Ryan et al. 2016; van Franeker and Bell 1988), as well as entanglement,
recorded for several marine mammal species (Waluda and Staniland 2013; Ivar
do Sul et al. 2011; Payne 1979). As plastic production and use are steadily
increasing, so will the concentration and impacts of this type of pollution in the
environment.

One characteristic of plastics in the ocean is their ability to host and
transport organisms among regions, which can potentially result in species
invasions (García-Gómez et al. 2021), including in the Antarctic Peninsula
(Barnes 2002). In marine systems, microbes quickly colonise plastics (Lobelle
and Cunliffe 2011), creating the “plastisphere” (Zettler et al. 2013). As a
consequence, ecological successions may occur, leading to mature communities
(Ramsperger et al. 2020; Kirstein et al. 2018) that can be composed of a wide
range of prokaryotes (Oberbeckmann and Labrenz 2020; Wright et al. 2020) and
eukaryotes (Davidov et al. 2020; Kettner et al. 2019). The term ‘plastisphere’,
initially used to describe microbial communities in marine systems, has been
expanded and now describes all organisms that live attached to plastics in
aquatic and terrestrial environments (Rillig et al. 2024; Du et al. 2022). Biofouling
on plastics can influence their weathering and contaminant absorbance (Rummel
et al. 2017), as well as in their vertical transport through the water column (Lobelle
et al. 2021; Rummel et al. 2017).

Although plastic pollution research is increasing in Antarctica (De-la-Torre
et al. 2024; Caruso et al. 2022), there has, to our knowledge, been no wide-scale
molecular surveys to describe the diversity of both prokaryotes and eukaryotes
of the plastisphere from plastics floating in the Southern Ocean. The long-term
monitoring studies developed by the High Latitudes Oceanography Group –
GOAL (Grupo de Oceanografia de Altas Latitudes, in Portuguese) in the last 20
62 years has shown increasing ice melting, and changes in phytoplankton
communities at the Antarctic Peninsula, mostly attributed to global warming
(Ferreira et al. 2020 and 2024; Mendes et al. 2023). It is therefore important to

gain baseline data on the composition of the plastisphere, so we can better
understand potential future changes in response to anthropogenic and climate
impacts.

In the early 2000s, Barnes and Fraser (2003) documented an assemblage
of animals attached to a piece of plastic that had washed ashore on Adelaide
Island in Antarctica, with at least 10 species spanning 5 different phyla. The
structure and function of prokaryotes in the Antarctic plastisphere has been
described in one study based on only two plastic items, one from land and one
from the sea (Cappello et al. 2021). Moreover, another study evaluated the
microbial biofilm colonising the surface of plastics (polyvinyl chloride - PVC, and
polyethylene - PE) in two coastal sites of the Ross Sea exposed to different
stressors (Caroppo et al. 2022). In addition, our research group has evaluated
plastics collected from the Western Antarctic Peninsula sampled in 2017
(Lacerda et al. 2019), describing the plastic-associated fungi using multiple DNA
markers (Lacerda et al. 2020).

To continue improving our understanding of the prokaryotic and
eukaryotic life associated with plastics in Antarctica on those samples, here we
used a multi-marker eDNA metabarcoding approach to characterise the full
diversity of life inhabiting these plastics. We investigated how these communities
varied between plastic shapes, polymeric composition and regional provinces
(northwestern and southwestern Antarctic Peninsula), and discussed their
ecological role in the region.

**Materials and Methods**

**Sampling area**

Plastics were sampled at the sea surface (water-air interface, around 15
91 cm depth) of 12 sampling stations, using a Manta net (330 µm mesh), during the
92 XXXVI Antarctic Operation and 7th expedition of project “Biological Interactions
in Marine Ecosystems off the Antarctic Peninsula Under Different Impacts of
Climate Change - INTERBIOTA”, in 2017 (Figure 1). The study area covered the
Gerlache Strait, which separates the Anvers and Brabant Islands from the

Commented [u2]: Please, update this section with recent papers on Antarctic marine plastisphere

Commented [u3R2]: The same applies to discussion

Commented [u4]: Please, add information about the hauling speed, duration of the sampling, and the distance covered during sampling, and so on

Antarctic Peninsula; the Bransfield Strait, between the southern Shetland Islands
and the Peninsula; and the north-eastern Bellingshausen Sea.

To better evaluate the effects of oceanographic features on the
plastisphere composition, the sampling stations were split into two data sets: the
Northwestern (NW) province, which includes the Bransfield Strait, and the
Southwestern (SW) province, which comprises the Gerlache strait and the region
under high influence of waters advected from the Bellingshausen Sea. These two
provinces present different bathymetry (Figure 1) and sea-ice coverage, which
are the major controllers of the biogeochemical spatial variability in the Southern
Ocean, making them distinct biogeochemical regions (Testa et al. 2021). At sub-
surface and deeper levels, the study region is influenced by water masses flowing
from the Bellingshausen and Weddell Seas, respectively called Transitional
Bellingshausen Water (TBW) and Transitional Weddell Water (TWW) (Kerr et al.
2018). The SW province, more influenced by the TBW, is characterised by a
regime of warmer, more nutrient-rich and more productive waters, while the NW
province has a greater influence of cold, more saline waters derived mainly from
the Weddell Sea shelf, and generally contains lower nutrient concentrations
(Holm-Hansen et al. 1997).

**Characterization of Plastics**

After each trawl, the contents of the collection cup were placed in an
aluminium bag and frozen at -20°C for posterior analysis of the plastisphere. In
the laboratory, samples were thawed separately and placed in a sterile container
filled with artificial sterile salt water (salinity 35) for manual separation of floating
plastic pieces (higher than 1mm in length) and biomass (Reisser et al. 2013).
Plastics pieces were picked up using sterile forceps, measured over their total
length and classified into categories according to their size (microplastic: < 5 mm,
and mesoplastic: 5-200 mm; adapted from Eriksen et al. 2014), shape (fragment,
foam, line, pellet and film) (GESAMP 2019), and polymer composition.

Each plastic piece was placed individually in a microcentrifuge tube with
absolute ethanol (reagent grade, MERK) to preserve the DNA of the associated
organisms, and 32 were submitted to genetic analysis. Polymer composition of
28 out of the 32 samples used for DNA analysis was determined through Fourier

Commented [u5]: Sterile?

Commented [u6]: How did you separate plastic pieces when frozen all together?

Commented [u7]: Biomass estimation?

Commented [u8R7]: It is not clear to me

Commented [u9]: Did you collect a total of 32 pieces of plastics?

Commented [u10R9]: If yes, this info should me moved to results

Commented [u11]: The same as above

128 Transform Infrared Spectroscopy (FTIR) with a SHIMADZU spectrometer, model
Prestige 21, using a diffuse reflectance module, 24 scans and 4 cm⁻¹ resolution.
FTIR procedures and data analysis follows the standard practice ASTM E1252-
98 (2013) (ASTM international 2013).

Figure 1. Sampling area of floating plastics and their associated plastisphere around the Western
Antarctic Peninsula. Sampling sites 1-6: Northwestern Province (NW); Sampling sites 7-12:
Southwestern Province (SW). This map was created with Python software by using the
"matplotlib" package.

Scanning Electron Microscopy (SEM)

A total of 14 plastic pieces (different from the ones submitted to DNA
extraction) were observed through Scanning Electron Microscopy (SEM) for the
detection of biofilms, aiming to investigate the morphology of the plastisphere
organisms. Plastics were initially dehydrated in absolute ethanol (Reagent-grade,
MERK), followed by fixation to an aluminium sheet with carbon tape and coated
with a 20–30 nm gold layer. The biofilm was observed using a JEOL microscope
(JSM 6610LV, JEOL, Tokyo), operated at 10–20 kV at a working distance of 10–
26 mm. Each item was imaged at different magnifications (20x to 40,000x) to
better identify the diversity of organisms. The main groups found on plastics were
qualitatively described.

Commented [u12]: Additional

148 **eDNA extraction, Amplification and Sequencing**

Plastic pieces were rinsed with sterile seawater before DNA extraction to
remove organisms that co-occurred with plastics during sampling. The total DNA
of plastic biofilms was extracted using a PowerSoil DNA extraction kit (Qiagen)
(Debeljak et al. 2017), with some modifications from the manufacturers'
instructions, as described in Lacerda *et al.* (2020). The quality and concentration
of extracted DNA were checked by spectrophotometry using Biodrop DUO
(Harvard Bioscience™). We then amplified the 16S V4, as well as the 18S V4
and V9 regions of the rRNA. Two regions of the 18S gene were used due to their
different resolutions on the diversity of eukaryotes (Choi and Park 2020); this was
confirmed in a recent study on the plastisphere from the South Atlantic Ocean
(Lacerda et al. 2022).

Primers 515f (5' - GTGYCAGCMGCCGCGGTAA - 3') and 806r (5' -
GGACTACNVGGGTWTCTAAT - 3') were used to amplify the 16S V4 region
(Walters et al. 2016); primers TAREuk454 (5' - CAGCASCYGC GGTAATTCC -
3') and TAREukRev3 (5' - ACTTTCGTTCTTGATYRA - 3') were used to amplify
the 18S V4 region (Stoeck et al. 2010); and primers 1391f (5'-
GTACACACCGCCCGTC-3') and EukB (5' -
TGATCCTTCTGCAGGTTACCTAC - 3') were used to amplify the 18S V9
region (Amaral-Zettler et al. 2009). PCR reactions and conditions for all molecular
markers are detailed in the Supplementary Material 1. Library preparation and
sequencing at the Illumina Mi-seq platform followed Lacerda et al. (2022 and
2020).

**DNA Sequence Processing and Data Analysis**

Sequences were analysed using a combination of software tools on
USEARCH v7.0.1090 (32Bit) (Edgar 2010) and QIIME v 1.8.0 (Caporaso et al.
2010). Sequence processing followed the steps: merging forward and reverse
reads, eliminating base spacers, separating primer sets, and applying quality
filters before converting data into FASTA files (Lacerda et al. 2020). These
FASTA files underwent dereplication, abundance sorting, and removal of
singleton sequences. Operational Taxonomic Units (OTUs) were formed via the
UPARSE clustering algorithm (Edgar 2013). Chimeras were removed using

UCHIME (Edgar et al. 2011), and an OTU table was generated by mapping OTUs
back to the original reads. OTUs containing fewer than three reads were excluded
from subsequent analyses.

16S sequences were classified at 98% similarity against the SILVA
database, while classification of both 18S V4 and V9 sequences was done at
97% similarity against the SILVA 132 database (Quast et al. 2013) using
UCLUST (Edgar 2010). Additionally, for taxonomy confirmation, sequences
underwent verification via the "Basic Local Alignment Search Tool (BLAST)"
against the comprehensive National Center for Biotechnology Information
(NCBI/Genbank) database. A review of relevant literature provided information
on the functional potential of prokaryotic communities. Refraction was performed
prior to statistical analysis and OTU tables were rarefied to 1,050 reads for the
16S marker, 400 reads for 18S V4, and 600 reads for 18S V9 marker.

Differences in alpha and beta diversity of plastic-associated organisms
(OTUs richness and community structure) among plastic categories (size, shape
and polymer composition) and by province were evaluated. An Analysis of
Variance (ANOVA) was performed to check for differences in OTUs richness per
plastic categories and provinces. The beta diversity was measured as the
average distance from the individual plastic to the category's median using Bray-
Curtis for the rRNA 16S dataset, and the binary Jaccard index for 18S V4 and V9
datasets.

To verify if differences in community structure could derive from within-
group variations, multivariate homogeneity tests of group dispersions
(PERMDISP) were conducted. Furthermore, a Permutational Multivariate
Analysis of Variance (PERMANOVA), with fixed factors and 9999 permutations,
was employed to assess potential disparities in community beta diversity among
categories (Anderson et al 2006). All statistical analyses were carried out using
the vegan packages (Oksanen et al. 2019) within R studio 1.1.456 (R Core
Team), and differences with $p \leq 0.05$ were deemed statistically significant.
Principal Coordinate Analysis (PCoA) was performed using the ggplot2 package
(Wickham 2009) in R studio to verify differences in the community composition
according to provinces and plastic categories, based on either Bray-Curtis
dissimilarity (prokaryotes) or Jaccard distance matrices (eukaryotes).

Results

Morphology of plastisphere organisms

SEM revealed different organisms living on floating plastics in Antarctica.
We observed diatoms, fungi and bacteria attached to plastic fragments (A, D, F),
lines (B, E) and foam (C) (Figure 5). Since the preservation method of samples
was focused on preserving the DNA (immediately frozen, followed by immersion
in absolute ethanol), some cells/structures could have broken, but it was still
possible to identify several groups living attached to plastics, reinforcing the
presence of taxa detected with the molecular data.

Commented [u13]: Was it applied to the 14 additional samples?

Commented [u14]: 2

Commented [u15]: (see below)

Figure 2. Scanning Electron Microscopy of organisms attached to floating plastics from the
Antarctic Peninsula. Centric (A) and pennate (B and C) diatoms; fungi (D and E) and bacteria cell
(F).

Commented [u16]: bacterial cells

DNA Sequence metrics

After quality filtering, the 16S rRNA gene dataset contained 299,615 reads
from 21 successful samples, comprising 1207 OTUs. Within the 16S dataset, the
number of OTUs per sample ranged from 59-474 (SE ± 26). For the eukaryotic
markers, 15 samples had the 18S V4 region successfully amplified, with 183,386
reads comprising 136 OTUs, whereas 26 samples had the 18S V9 marker
amplified, with 181,395 reads comprising 415 OTUs. The number of OTUs per

Commented [u17]: (out of ?)

Commented [u18]: from 59 to 474

sample ranged from 4-69 (SE ± 4.15) and 12-124 (SE ± 6.2) in the 18S V4 and
 V9 datasets, respectively.

**Prokaryotic diversity in the Antarctic plastisphere**

Within the 16S dataset we detected 32 Phyla of Bacteria and two Archaea,
 as well as a number of “unclassified bacteria” (Figure 3). The most abundant
 phyla within the dataset were Proteobacteria (Relative abundance (RA) of 44%)
 and Bacteroidetes (RA 27%) (Figure 3). Among Proteobacteria groups,
 Gammaproteobacteria was composed of 202 OTUs, while Alpha and
 Deltaproteobacteria contained 153 and 54 OTUs, respectively.

At family level, the most abundant bacterial group was Flavobacteriaceae
 (Bacteroidetes, comprised of 63 OTUs, and with relative abundance of 19%), and
 Rhodobacteraceae (Proteobacteria, comprised of 48 OTUs, and relative
 abundance of 13%). The Archaea group was composed by phyla
 Thaumarchaeota and Euryarchaeota, and was neither among the most abundant
 nor the most frequent in the 16S dataset. The most abundant archaea OTU was
 a *Methanosarcina* sp. (RA 0.3%, FO 9.5%), whereas the most frequent archaea
 was *Nitrososphaeraceae* sp. (FO 28%, RA 0.004%).

Commented [u19]: phyla
 Commented [u20]: two of Archaea
 Commented [u21]: relative

Commented [u22]: Family?
 Commented [u23]: Please, check for English

 Figure 3. Relative abundance (%) of prokaryotic phyla per plastic sample identified through the
 rRNA 16S gene according to the province (NW: Northwestern; SW: Southwestern) of the Antarctic
 peninsula.

The most abundant Bacteria OTU (5% of relative abundance) matched
with 100% of confidence many uncultured Bacteroidales sequences from NCBI.
The second most abundant OTU was classified as *Octadecabacter antarcticus*,
and represented 4% of relative abundance; these two OTUs were present in over
half of the samples, with 67% and 76% of frequency of occurrence (FO%),
respectively. In addition, among the ten most abundant prokaryotic OTUs, we
found many uncultured species, such as uncultured Terasakiellaceae (RA 4%,
FO 65%), uncultured Acinetobacter (RA 3%, FO 73%), as well as uncultured
Rhodobacteraceae and *Alkalibacterium* sp. (both with RA 2%, and FOs of
respectively 81% and 62%). Moreover, two *Polaribacter* species were also
among the ten most abundant OTUs, one classified as *Polaribacter* sp. *MB-G19*
and the other as uncultured *Polaribacter* sp., both being frequent in more than
70% of samples and showing relative abundances of 3% and 2%, respectively.

**Functional potential of prokaryotes in the Antarctic plastisphere**

The Antarctic plastisphere hosts bacterial groups that have been
previously described to potentially degrade hydrocarbons and plastics, such as
*Tenacibaculum* spp., *Oleispira* sp., *Pseudomonas* spp., *Acidovorax* sp.,
*Comamonas* sp., *Ralstonia* sp., *Schlegelella* sp., *Alcanivorax* sp., as well as
species belonging to the genera *Vibrio* and *Bacillus*. We also found prokaryotic
groups that have been previously reported as potential pathogens, such as *Vibrio*
and *Acinetobacter* spp.

**Eukaryotic diversity in the Antarctic plastisphere**

The two eukaryotic datasets detected slightly different taxonomic groups.
Rotifera, Gastrotricha and Perkinsidae (comprised into "Other Alveolata") were
identified only by the 18S-V4, whereas 13 other groups were exclusive to the
18S-V9 dataset: Retaria, Tunicata, Chaetognatha, Haptophyta, Excavata,
Porifera, Apicomplexa, Apusomonadidae, Centrohelida, Picozoa, Telonema,
Bryozoa and Charophyta. Twenty-one other eukaryotic groups were identified by
both molecular markers: Fungi, Diatom, Dinoflagellata, Chlorophyta, Ciliophora,
Cnidaria, Cercozoa, Phaeophyceae, Rhodophyta, Choanoflagellida,
Amoebozoa, Annelida, Chrysophyceae, Crustacea, Ctenophora, Telonema,
Nematoda, Cryptophyta, Mollusca and Platyhelminthes, along with

Commented [u24]: In my opinion, as the authors did not carry a functional metabolic profiling, this text (being a comment) should be moved in the discussion

uncultured/unidentified groups, as well as “Other Alveolata” and “Other
 Stramenopiles” (Figure 4).

 Figure 4. Percentage of eukaryotes frequency of occurrence (FO) of eukaryotic groups from the
 marine Antarctic plastisphere, identified through the 18S gene (V4 and V9 regions), separated by
 province (North-western and South-western). For better visualisation, only groups with FO higher
 than 15% are shown.

Fungi was observed in 100% of samples within the 18S-V9. Within the
 18S-V4 dataset, there was no group present in all samples, with the most
 frequent eukaryotes identified by the 18S-V4 marker being Dinoflagellata, with
 FO of 87%. Other groups such as Diatom and Ciliophora were highly frequent in
 samples from both datasets. Many diatom OTUs matched species previously

described in NCBI for the Southern Ocean/polar regions, such as the benthic
diatom *Navicula glaciei* (Genbank access number EF106789.1), *Chaetoceros*
*socialis* (KX253957.1) and *Corethron inerme* (AJ535180.1). The 18S-V9 marker
dataset showed many “unidentified/uncultured” eukaryotes (FO 81%), whereas
this was only observed with 7% of FO in the 18S-V4 dataset.

Although there was a greater diversity of eukaryotes, only a few groups
were frequent in more than 50% of samples, with the remaining showing lower
occurrence on plastics (Figure 4). When considering the most frequent eukaryotic
OTUs, *Symbiodinium* dinoflagellate (symbionts of Radiolaria), as well as
Chlorophyta *Klebsormidium* and *Monostroma* spp., and the Diatom *Thalassiosira*
sp. stood out in the 18S-V4 dataset. Meanwhile, for the 18S-V9 dataset, fungal
species from genera *Aspergillus* and *Sterigmatomyces*, Diatom *Fragilaria* sp.,
and the uncultured Ciliophora *Spirotontonia* sp. were the most frequent OTUs.
We also identified microeukaryotes described as animal parasites, such as the
ciliate *Uronema* sp. Potential harmful fungi associated with Antarctic plastics are
not reported here, since they were described in detail in Lacerda *et al.* (2020).

**Species richness and community structure according to provinces and** 316 **plastic categories**

The prokaryotic species richness (number of OTUs) was not significantly
variable according to provinces or plastic categories (shape and polymer)
(ANOVA, province: p-value = 0.58; shape: p-value = 0.71; polymer: p-value =
0.27) (Figure 5). However, the community structure of prokaryotes was different
between plastics from the Northwestern and Southwestern provinces
(PERMANOVA; F = 1.5222, p-value = 0.05) (Figures 3 and 5), although it did not
vary among any of the different plastic categories. No interaction of province and
plastic categories significantly influenced the prokaryotic community composition
(p-value = 0.8806). In the NW province, Bacteroidetes and Verrucomicrobia were
more abundant, while in the SW province Acidobacteria, Chloroflexi,
Sinergystetes and Epsilonbacteraeota were the most representative groups.

**Figure 5.** Mean number of observed OTUs per plastic sample from the Antarctic Peninsula,
 according to geographic province (Northwestern and Southwestern), polymer type (PA -
 polyamide; PE - polyethylene; PP - polypropylene; PS - polystyrene; PU - polyurethane) and
 plastic shape (foam, fragment, line and pellet). Data was obtained from rarefied 16S (1,000
 sequence/sample), 18S-V4 (300 sequences/sample) and 18S-V9 (500 sequences/sample)
 amplicon sequence libraries.

For eukaryotes, OTUs richness did not vary according to any category of
 plastics or geographic provinces (ANOVA, for the 18S-V4 dataset - province: p-

Commented [u25]: The authors should consider to cite "province", Polymer and shape only one time per graph typology

value = 0.36; shape: p-value = 0.58; polymer: p-value = 0.83. For the 18S-V9
dataset - province: p-value = 0.29; shape: p-value = 0.94; polymer: p-value =
0.82), but as for prokaryotes, the community structure was different between the
Northwestern and Southwestern provinces for the 18S-V9 dataset
(PERMANOVA, $F = 1.4429$, $p = 0.007$) (Figure 6), which identified more taxa than
the 18S-V4. The combination of factors (province and plastic categories) did not
influence the diversity or community structure of the Antarctic plastisphere
(PERMANOVA, 18S-V4: p-value = 0.27; 18S-V9: p-value = 0.67).

Figure 6. Non-metric multidimensional scaling (NMDS) plots of the marine plastisphere from the
Western Antarctic peninsula based on Bray-Curtis distance matrix for rRNA 16S, and on Jaccard
distance matrix for 18S-V4 and 18S-V9, according to province (NW: Northwestern and SW:
Southwestern).

Discussion

This study is the first comprehensive survey within Antarctica of the marine
plastisphere, assessing both species richness and community composition, and
is one of the few studies within the Southern hemisphere (Audrézet et al. 2022;
Agostini et al. 2021; Lacerda et al. 2020). We show a wide range of prokaryotes
and eukaryotes living on plastics in Antarctica, with community composition being
shaped only by provinces and not by plastic categories. This is corroborated by
the pattern observed for the plastisphere of several other areas, in which the
geographic location influences plastisphere composition more than the
characteristics of the plastics themselves.

The Antarctic plastisphere compared to other regions

In accordance with several marine plastisphere descriptions for other
regions (Reisoglu et al. 2024; Sérvulo et al. 2023; Lacerda et al. 2022; Lee & Park
2022), we detected many benthic species (e.g. the Diatoms *Navicula* sp. and
*Nitzschia* sp.), which are likely well adapted to a life attached to substrates. As
Antarctica does not have any large trees or shrubs, the majority of debris entering
marine systems is either man-made or natural debris carried to the region by
ocean currents or direct local disposal (Lacerda et al. 2019). Our rRNA 16S
results are similar to what has been observed for other regions (see Wright et al.
2021 for a review of studies from the northern hemisphere), with a dominance of
Proteobacteria and Bacteroidetes, specifically Alpha and Gammaproteobacteria
(Sun et al. 2023). The majority of prokaryotes and eukaryotes that we observed
on plastics are common taxa from sediments (Currie et al. 2020), snow (Soto et
al. 2023), and seawater in Antarctica (Fonseca et al. 2022). Here, we have shown
that these organisms are currently interacting with this new anthropogenic
stressor in the Southern Ocean.

Archaea occurred very rarely in the Antarctic plastisphere, which was also
the case in the Baltic Sea (Kettner et al. 2019), and areas in the North Pacific
(Bryant et al. 2016) and South Atlantic Ocean basins (Lacerda et al. 2022,
Agostini et al. 2021). The dominance of bacteria in our samples, as well as those
from around the world (Wright et al. 2020), is due to the fact that they are one of
the most abundant organisms in marine biofilms, besides being highly
interconnected with eukaryotes on plastics (Amaral Zettler et al. 2020; Kettner et
al. 2019). The dominance of eukaryotes from the SAR supergroup
(Stramenopiles, Alveolata & Rhizaria) and Fungi in the Antarctic plastisphere is
aligned with other regions around the globe. For example, when analysing the V4
region of the 18S gene, Kettner *et al.* (2019) found that most of the
microeukaryotes living on plastics off the coast of Germany were from the SAR
supergroup, Fungi, Metazoa, and Chloroplastida. In addition, these groups were
also frequent in the plastisphere from the South Pacific (Audrézet et al. 2022) and
South Atlantic (Sérvulo et al. 2023; Lacerda et al. 2022) ocean basins. Our
findings demonstrate that the Antarctic plastisphere harbours a similar microbial
community to other global regions. However, unique Antarctic taxa were also

present, indicating a localised adaptation of some taxa to this emerging
environmental stressor.

**Diversity across Antarctic marine ecosystems**

The diversity of marine life in Antarctica is a relatively understudied
subject, particularly when it comes to the use of advanced multi-marker DNA
metabarcoding techniques (Angulo-Preckler et al. 2023). Due to the high rate of
endemism, Antarctica is particularly vulnerable to disturbances, and species loss
in the Southern Ocean is more likely to be a loss of global biodiversity (Barnes &
Peck 2008). Therefore, plastics in the ocean, which can serve as substrates for
various benthic and planktonic species, pose a significant threat to these
systems. The attachment of diverse taxa to plastic debris can alter natural
processes by facilitating the transport of these organisms; additionally, the
plastisphere can be a self-sustaining system, allowing many ecological relations
to take place (Delacuvellerie et al. 2022; Amaral-Zettler et al. 2020), and thus
increasing the chances of survival of several groups.

Many studies, such as those conducted by Fonseca et al. (2022), have
identified dominant prokaryotic taxa like Proteobacteria, Bacteroidetes, and
Planctomycetes in Antarctic benthic communities, which correspond closely with
those found in our plastic samples. In addition, they also found various benthic
eukaryotic organisms, including Nematoda, Arthropoda, and Mollusca groups,
which were also observed in our plastisphere samples. Moreover, Angulo-
Preckler *et al.* (2023) identified a great diversity of Arthropoda, Bacillariophyta,
and Annelida in shallow hard-bottom communities from the Western Antarctic
Peninsula, which are in accordance with our results for plastics. This reinforces
the role of plastics in hosting benthic species that can now be found floating at
the sea surface after attaching to plastics.

These recent works also point out significant differences in microbial
communities across various locations along the Western Antarctic Peninsula,
suggesting high levels of endemism and variability in community composition
(Angulo-Preckler et al. 2023; Fonseca et al. 2022). However, the transport of
plastics across different regions threatens these natural boundaries, as floating

plastics can introduce non-native species, which could lead to further ecological
imbalances.

Regarding the free-living organisms around the Antarctic Peninsula, Luria
*et al.* (2016) observed that bacterial community composition shifted seasonally,
with increased abundance of several taxa associated with phytoplankton blooms
during summer, particularly *Polaribacter* species. Indeed, two *Polaribacter* OTUs
were among the ten most abundant bacteria we observed within the Antarctic
plastisphere. Moreover, Flavobacteriaceae and Rhodobacteraceae were
abundant free-living bacterial groups during early summer, but their populations
declined in February and March (Luria *et al.* 2016). However, in our plastisphere
samples collected in mid-late February, these families were the most abundant.
This raise concerns that plastics may create unique microenvironments, allowing
certain species to thrive under conditions that differ from natural water
ecosystems. Although research on polar ecosystems is expanding, recent
reviews have largely overlooked the impact of plastics and the plastisphere,
highlighting the need for more comprehensive studies in this field.

**Provinces drive community composition**

Although some studies have shown differences in the plastisphere
community composition according to polymer type, most studies highlight that the
polymer does not seem to influence the diversity and community structure of the
plastisphere (Amaral-Zettler *et al.* 2020). This seems to be especially true for
mature communities formed on plastics that may have been in the environment
for long periods of time (Wright *et al.* 2021). Here we evaluated plastics collected
from the environment and observed that, indeed, polymer type did not shape the
plastisphere in Antarctica. Moreover, a recent study reported no correlation
between any polymer and biofouling type on plastics washed up on the shore in
the Antarctic Specially Protected Area Robert Island (Johnston *et al.* 2024).

Basili *et al.* (2020) stated that geographical location, rather than polymer,
shapes the plastisphere community structure. In alignment with this statement,
we observed differences among the Northwestern and Southwestern provinces
of the Antarctic Peninsula, which could be explained by the local environmental
factors such as ocean circulation, sea surface temperature and macronutrients.

When describing the hydrographic properties and macronutrients variability in the
Southern Ocean based on a time series spanning from 1996 to 2019, Monteiro
*et al.* (2023) observed warmer waters in the Gerlache strait (higher than 0 °C),
whereas the Bransfield strait presented colder surface waters (< 0 °C). In
addition, seasonal macronutrient drawdown for the same year we conducted
sampling (2017) showed higher values for dissolved inorganic nitrogen (DIN),
phosphate and silicic acid in the Gerlache strait (Monteiro *et al.* 2023). These
authors also showed that the silicic acid/N (Si:N) uptake ratio during the austral
summer in 2017 was higher in the Bransfield strait than in the Gerlache strait.
Such results highlight the biogeochemical differences among our two sampled
provinces, reinforcing that the environmental factors might have an influence on
microbial communities (Signori *et al.* 2014), including the ones living on plastics.

Commented [u26]: waters

We observed a higher diversity of prokaryotic and eukaryotic taxa in the
SW province (Gerlache strait and Bellingshausen Sea), compared to the NW
province (Bransfield strait). This could be explained by the warmer waters present
in the SW province when compared to the colder and more saline waters of the
northern part of the Bransfield Strait (Sangrà *et al.* 2011), where most sampling
stations in the NW province were located. At this region (NW province), Signori
*et al.* (2014) observed that deep water samples were significantly enriched by
Archaea, Plantomycetes and Chloroflexi, whereas shallow water samples
showed a higher contribution of Bacteroidetes, and Proteobacteria (mostly
Rhodobacteraceae and Oceanospirillaceae).

We also observed Bacteroidetes and Proteobacteria as the dominant
groups in samples from the NW province, but the aforementioned groups
previously found in deep waters in the NW province had greater abundances on
floating plastics from the SW province. The entrance of the Bransfield current on
the northwestern Antarctic peninsula, along with small-scale vortexes, may
structure the plastisphere communities. However, strong events such as storms,
especially during the winter season, may overcome these barriers and allow
plastics - with their associated communities - to travel around the Antarctic
continent, as demonstrated in a 7-year plastics dispersal model in Antarctica
(Lacerda *et al.* 2019).

**Environmental implications of the plastisphere in Antarctica**

The marine plastisphere in Antarctica presents significant ecological
implications due to its ability to alter natural ecological interactions. Plastics in
marine environments, especially in remote regions like Antarctica, serve as
vectors for the dispersal of various organisms that might not naturally reach these
areas. These communities can disrupt local ecosystems by introducing non-
native species (García-Gómez et al. 2021) altering food webs (Tuuri & Leterme
2023), and impacting biogeochemical cycles (Galvani & Loiselle 2021). For
example, the pennate diatom *Navicula glaciei*, typically ice-associated, was found
adhering to our plastic samples, as identified by DNA analysis in this study and
by scanning electron microscopy in Lacerda et al. (2019). This indicates a
potential for increased dispersal and ecological shifts in areas where it would not
naturally thrive.

The plastisphere can also enhance the accumulation of pollutants on
plastic debris (Lenoble et al. 2024), which exacerbates the ecotoxicological
impacts on marine life. Biofilm-covered plastics can mimic the appearance and
smell like natural food, leading to an increased ingestion of plastics by marine
animals (Amaral-Zettler et al. 2015). The ingestion of these plastics by Antarctic
species could lead to the transfer of harmful pollutants and pathogens, posing a
threat to the fragile ecosystems of the region.

Climate change is likely to further exacerbate the ecological impacts of the
Antarctic microbial communities (Santos et al. 2023), which can include the
plastisphere. Rising temperatures and changes in ocean chemistry could
influence the composition and function of plastisphere communities (Kerfahi et al.
2023; Harvey et al. 2020), as evidenced by studies in other regions (Nguyen et
al. 2023; Ji et al. 2022, Pinnell & Turner 2020). In warmer conditions, such as
those observed during anomalously warm summers in Antarctica, the
biodegradation of plastics may increase, potentially releasing even more
pollutants and altering microbial communities. Bacteria species with potential to
degrade plastics, such as *Oleispira* sp., *Alcanivorax* sp., Colwelliaceae and
Vibrionaceae spp., were indeed found in our samples. Furthermore, the increase
in temperature may favour the virulence of pathogens, such as *Vibrio* species

Commented [u27]: ;

(Billaud et al. 2022). This underscores the urgent need for further research to
understand the long-term implications of the plastisphere in Antarctica.

**Conclusion**

Using a combination of DNA-metabarcoding and scanning electron
microscopy, we provided a detailed and high-resolution characterization of the
marine plastisphere in Antarctica, revealing a diverse range of prokaryotic and
eukaryotic organisms. Given that polar regions typically exhibit lower biodiversity
and fewer trophic levels, our findings highlight the role of plastics as a new,
mobile, and durable substrate that, in addition to other anthropogenic stressors,
has the potential to significantly alter the local ecosystems. We demonstrated that
many benthic/sessile groups from coastal areas are now living attached to
floating plastics in the open ocean in the region. Moreover, several
microorganisms within the marine plastisphere possess biotechnological
potential, emphasising the need for omics-based approaches to further explore
and understand the ecological functions of the Antarctic plastisphere. As plastic
pollution increases in the region, we can expect a greater availability of artificial
substrates for the establishment of diverse species. One of the key
recommendations of this study is to include comparative analyses of plastisphere
communities between the eastern (Weddell Sea) and western Antarctic
Peninsula, as this is crucial for understanding the differential impacts of climate
change on biota in these distinct areas. Key areas for future research also include
understanding the limits of species coexistence on plastic surfaces and the
impacts of these communities on other species in the Southern Ocean.
Addressing these questions will be vital for predicting and mitigating the long-
term ecological consequences of plastic pollution in Antarctica.

**Competing Interests**

The authors declare no competing interests.

**Data availability Statement**

The DNA sequencing data that support the findings of this study are available
from the European Nucleotide Archive (ENA) (link will be available upon

acceptance). Any additional data that support the conclusions of this study are
available from the corresponding author upon request.

**Author Contributions**

**ALL** study conception, sampling, laboratory work and data analysis; **MP** study
conception and sampling; **ES** and **CRM** obtained funding and conceived the
oceanographic survey design; **FK** performed FTIR analysis. **JT** conducted
laboratory work and data analysis. **ALL** wrote the first draft of the paper, and all
authors contributed to discussing and editing the manuscript.

**Funding**

This study was conducted within the activities of project INTERBIOTA, financed
by the Conselho Nacional de Desenvolvimento Científico e Tecnológico - CNPq
Grant number 407889/2013-2. CNPq also provided research fellowship to ALdFL
(SWE 206250/2017-7), MCP (PQ 312470/2018-5), FK (PQ 435612/2018-2), and
ERS (PQ 310597/2018- 8) during the development of this study. ALdFL received
a scholarship from the Coordination for the Improvement of Higher Education
Personnel (Coordenação de Aperfeiçoamento de Pessoal de Nível Superior –
CAPES), which also provided access to the Portal de Periódicos and financial
support through Programa de Excelência Acadêmica – PROEX. JDT was funded
by a UKRI NERC grant NE/X012204/1.

**Acknowledgment**

This work is a contribution of the High Latitude Oceanography Group (GOAL)
under the scope of the Brazilian Antarctic Program (PROANTAR). We thank
Carlos Fujita and the crew of the NPo Almirante Maximiano for helping with
oceanographic survey and sampling. We thank Dr Thorunn Helgason, Dr Sally
James and Dr Peter Aston from the University of York Genomics & Bioinformatics
Laboratory, for assistance, support and technical expertise with sequencing. We
thank Dr. Lucas R. Almeida for producing the map in Figure 1 of the manuscript.
We thank Mikaele, Lijainah and Nathalia for their assistance with FTIR analysis.
SEM images were taken at the Electron Microscopy Center (CEME-Sul,
PROPESP-FURG) with the assistance of Rudmar Krumreick and Caroline Ruas.

**References**

- Agostini L. et al. Deep-sea plastisphere: long-term colonization by plastic-
associated bacterial and archaeal communities in the Southwest Atlantic
Ocean. *Sci. Total. Environ.* **793**: 148335. 10.1016/j.scitotenv.2021.148335
(2021).
- Amaral-Zettler LA, EA McCliment, HW Duckow, SM Huse. A Method for
Studying Protistan Diversity Using Massively Parallel Sequencing of V9
Hypervariable Regions of Small-Subunit Ribosomal RNA Genes. *PLoS ONE*
**4**(7): e6372. 10.1371/journal.pone.0006372 (2009)
- Amaral-Zettler LA, et al. The biogeography of the Plastisphere: implications
for policy. *Front. Ecol. Environ.* **13**(10), 541–546. 10.1890/150017 (2015).
- Amaral-Zettler LA, ER Zettler, TJ Mincer. Ecology of the Plastisphere. *Nat.*
*Rev. Microbiol.* **18**, 139-151. 10.1038/s41579-019-0308-0 (2020).
- Anderson MJ, KE Ellingsen, BH McArdle. Multivariate Dispersion as a
Measure of Beta Diversity. *Ecol. Lett.* **9**(6), 683–693. 10.1111/j.1461-
0248.2006.00926.x (2006).
- Angulo-Preckler C, M Turon, K Præbel, C Avila, C, OS Wangensteen.
Spatio-temporal patterns of eukaryotic biodiversity in shallow hard-bottom
communities from the West Antarctic Peninsula revealed by DNA
metabarcoding. *Divers. Distrib.* **29**, 892–911. [10.1111/ddi.13703](https://doi.org/10.1111/ddi.13703) (2023).
- ASTM E1252. Standard Practice for General Techniques for Obtaining
Infrared Spectra for Qualitative Analysis. *Annual Book of ASTM Standards*
**3**, 1–13. doi:10.1520/E1252-98R13.2 (2013).
- Audrézet F. et al. Eco-Plastics in the Sea: Succession of Micro- and Macro-
Fouling on a Biodegradable Polymer Augmented with Oyster Shell. *Front.*
*Mar. Sci.* **9**: 891183. <https://doi.org/10.3389/fmars.2022.891183> (2022).
- Barnes DKA. Invasions by Marine Life on Plastic Debris. *Nature*, **416**, 808–
809. 10.1038/416808a (2002).

Barnes DK & KP Fraser. Rafting by five phyla on man-made flotsam in the
Southern Ocean. *Mar. Ecol. Prog. Ser.* **262**, 289–291 (2003).

Barnes DKA, LS Peck LS. Vulnerability of Antarctic shelf biodiversity to
predicted regional warming. *Clim. Res.* **37**, 149–163. [10.3354/cr00760](https://doi.org/10.3354/cr00760)
(2008).

Basili M, *et al.* Major role of surrounding environment in shaping biofilm
community composition on marine plastic debris. *Front. Mar. Sci.* **7**: 262.
[10.3389/fmars.2020.00262](https://doi.org/10.3389/fmars.2020.00262) (2020).

Billaud M., F Seneca, E Tambutte, D Czerucka. An increase of seawater
temperature upregulates the expression of *Vibrio parahaemolyticus*
virulence factors implicated in adhesion and biofilm formation. *Front.*
*Microbiol.* **13**, 1–10. [10.3389/fmicb.2022.840628](https://doi.org/10.3389/fmicb.2022.840628) (2022).

Bryant JA, *et al.* Diversity and activity of communities inhabiting plastic debris
in the North Pacific Gyre. *MSystems* **1**(3): e00024-16.
[10.1128/mSystems.00024-16](https://doi.org/10.1128/mSystems.00024-16) (2016).

Caporaso JG, *et al.* QIIME Allows Analysis of High- Throughput Community
Sequencing Data. *Nat. Methods* **7**(5), 335–336. 10.1038/nmeth0510-335
(2010).

Cappello S, *et al.* New Insights into the Structure and Function of the
Prokaryotic Communities Colonizing Plastic Debris Collected in King George
Island (Antarctica): Preliminary Observations from Two Plastic Fragments.
*J. Hazard. Mater.* **414**: 125586. 10.1016/j.jhazmat.2021.125586 (2021).

Caroppo C, *et al.* Microbial Biofilms Colonizing Plastic Substrates in the Ross
Sea (Antarctica). *J. Mar. Sci. Eng.* **10**: 1714.10.3390/jmse10111714 (2022)

Caruso G, E Bergami, N Singh, I Corsi. Plastic occurrence, sources, and
impacts in Antarctic environment and biota. *Water Biol. Secur.* **1**(2): 100034.
[10.1016/j.watbs.2022.100034](https://doi.org/10.1016/j.watbs.2022.100034) (2022).

Choi J & JS Park. Comparative Analyses of the V4 and V9 Regions of 18S
rDNA for the Extant Eukaryotic Community Using the Illumina Platform. *Sci.*
*Rep.* **10**: 6519. 10.1038/s41598-020-63561-z (2020).

Cunningham EM, *et al.* High Abundances of Microplastic Pollution in Deep-
Sea Sediments: Evidence from Antarctica and the Southern Ocean. *Environ.*
*Sci. Technol.* **54** (21), 13661–71. 10.1021/acs.est.0c03441 (2020).

Currie AA, *et al.* Sea ice dynamics drive benthic microbial communities in
McMurdo Sound, Antarctica. *Front. Microbiol.* **12**: 745915.
10.3389/fmicb.2021.745915 (2021).

Davidov K, *et al.* Identification of Plastic-Associated Species in the
Mediterranean Sea Using DNA Metabarcoding with Nanopore MinION. *Sci.*
*Rep.* **10**: 17533. 10.1038/s41598-020-74180-z (2020).

De-la-Torre GE, *et al.* Assessing the current state of plastic pollution
research in Antarctica: Knowledge gaps and recommendations.
*Chemosphere* **355**: 141870. 10.1016/j.chemosphere.2024.141870 (2024).

Debeljak P, *et al.* Extracting DNA from Ocean Microplastics: A Method
Comparison Study. *Anal. Methods* **9**(9), 1521–1523. 10.1039/c6ay03119f
(2017).

Delacuvellerie A., *et al.* From rivers to marine environments: A constantly
evolving microbial community within the plastisphere. *Marine Pollution*
*Bulletin*, **179**: 113660. [10.1016/j.marpolbul.2022.113660](https://doi.org/10.1016/j.marpolbul.2022.113660) (2022).

Du, Y, X Liu, X Dong, Z Yin. A review on marine plastisphere: biodiversity,
formation, and role in degradation. *Comput. Struct. Biotechnol.* **20**, 975–988.
[10.1016/j.csbj.2022.02.008](https://doi.org/10.1016/j.csbj.2022.02.008) (2022).

Edgar RC. Search and Clustering Orders of Magnitude Faster than BLAST.
*Bioinformatics* **26**, 2460–2461. 10.1093/bioinformatics/btq461 (2010).

Edgar RC, BJ Haas, JC Clemente, C Quince, R Knight. UCHIME Improves
Sensitivity and Speed of Chimera Detection. *Bioinformatics* **27**, 2194–2200.
10.1093/bioinformatics/btr381 (2011).

Edgar RC. UPARSE: Highly Accurate OTU Sequences from Microbial
Amplicon Reads. *Nat. Methods* **10**, 996–998. 10.1038/nmeth.2604 (2013).

Eriksen M, *et al.* Plastic Pollution in the World's Oceans: More than 5 Trillion
Plastic Pieces Weighing over 250,000 Tons Afloat at Sea. *PLoS ONE* **9**(12):
e111913. 10.1371/journal.pone.0111913 (2014).

Ferreira A., *et al.* Changes in phytoplankton communities along the Northern
Antarctic Peninsula: causes, impacts and research priorities. *Front. Mar. Sci.*
**7**: 576254. <https://doi.org/10.3389/fmars.2020.576254> (2020).

Ferreira A., *et al.* 2024. Climate change is associated with higher
phytoplankton biomass and longer blooms in the West Antarctic Peninsula.
*Nat. Commun.* **15**:6536. [10.1038/s41467-024-50381-2](https://doi.org/10.1038/s41467-024-50381-2).

Fonseca VG, *et al.* Metabarcoding the Antarctic Peninsula biodiversity using
a multi-gene approach. *ISME commun.* **2**, 37. 10.1038/s43705-022-00118-3
(2022).

Galgani L & SA Loiselle. Plastic pollution impacts on marine carbon
biogeochemistry. *Environ. Pollut.* **268** : 115598.
10.1016/j.envpol.2020.115598 (2021).

García-Gómez JC, M Garrigós, J Garrigós. Plastic as a Vector of Dispersion
for Marine Species with Invasive Potential: A Review. *Front. Ecol. Evol.*, **9**:
629756. 10.3389/fevo.2021.629756 (2021).

GESAMP. Guidelines for the Monitoring and Assessment of Plastic Litter in
the Ocean. *GESAMP Reports & Studies* **99**: 130p.
[http://www.gesamp.org/publications/guidelines-for-the-monitoring-and-](http://www.gesamp.org/publications/guidelines-for-the-monitoring-and-assessment-of-plastic-litter-in-the-ocean)
[assessment-of-plastic-litter-in-the-ocean](http://www.gesamp.org/publications/guidelines-for-the-monitoring-and-assessment-of-plastic-litter-in-the-ocean) (2019).

Harvey BP, *et al.* Ocean acidification alters bacterial communities on marine
plastic debris. *Mar. Pollut. Bull.* **161**: 111749.
10.1016/j.marpolbul.2020.111749 (2020).

Holm-Hansen O, *et al.* Distribution of phytoplankton and nutrients in relation
to different water masses in the area around Elephant Island, Antarctica.
*Polar Biol.* **18**, 145–153. 10.1007/s003000050169 (1997).

Isobe A, K Uchiyama-Matsumoto, K Uchida, T Tokai. Microplastics in the
Southern Ocean. *Mar. Pollut. Bull.* **114**, 623–626.
10.1016/j.marpolbul.2016.09.037 (2017).

Ivar do Sul, JA, *et al.* Plastics in the Antarctic Environment: are we looking
only the tip of the iceberg? *Oecol. Aust.* **15**, 150–170.
10.4257/oeco.2011.1501.11 (2011).

Ji L, B Tanunchai, SFM Wahdan, M Schädler, W Purahong. Future climate
change enhances the complexity of plastisphere microbial co-occurrence
networks, but does not significantly affect the community assembly. *Sci.*
*Total Environ.* **844**, 157016. 10.1016/j.scitotenv.2022.157016 (2022).

Johnston LW, C Manno, CX Salinas. Assessment of plastic debris and
biofouling in a specially protected area of the Antarctic Peninsula region.
*Mar. Pollut. Bull.* **207**: 116844. 10.1016/j.marpolbul.2024.116844 (2024).

Kelly A, D Lannuzel, T Rodemann, KM Meiners, HJ Auman. Microplastic
Contamination in East Antarctic Sea Ice. *Mar. Pol. Bull.* **154**: 111130.
10.1016/j.marpolbul.2020.111130 (2020).

Kerfahi D, *et al.* Whole community and functional gene changes of biofilms
on marine plastic debris in response to ocean acidification. *Microb. Ecol.* **85**,
1202–1214. [10.1007/s00248-022-01987-w](https://doi.org/10.1007/s00248-022-01987-w) (2023).

Kerr R, MM Mata, CRB Mendes, ER Secchi. Northern Antarctic Peninsula:
a marine climate hotspot of rapid changes on ecosystems and ocean
dynamics. *Deep Sea Res. Part II Top. Stud. Oceanogr.* **149**, 4–9.
10.1016/j.dsr2.2018.05.006 (2018).

Kettner MT, S Oberbeckmann, M Labrenz, HP Grossart. The Eukaryotic Life
on Microplastics in Brackish Ecosystems. *Front. Mar. Sci.* **10**: 538.
10.3389/fmicb.2019.00538 (2019).

Kirstein, IV, A Wichels, G Krohne, G Gerds. Mature Biofilm Communities on
Synthetic Polymers in Seawater - Specific or General?. *Mar. Environ. Res.*
**142**: 147–154. 10.1016/j.marenvres.2018.09.028 (2018).

Lacerda ALDF, *et al.* Plastics in Sea Surface Waters around the Antarctic
Peninsula. *Sci Rep* **9**: 3977. 10.1038/s41598-019-40311-4 (2019).

Lacerda ALDF, MC Proietti, ER Secchi, JD Taylor. Diverse Groups of Fungi
Are Associated with Plastics in the Surface Waters of the Western South
Atlantic and the Antarctic Peninsula. *Mol. Ecol.* **29**: 1903–1918.
10.1111/mec.15444 (2020).

Lacerda AL, *et al.* Floating plastics and their associated biota in the Western
South Atlantic. *Sci. Tot. Environ.* **805**: 150186.
10.1016/j.scitotenv.2021.150186 (2022).

Lee B & MG Park. Drifting marine plastics as new ecological habitats for
harmful eukaryotic microbial communities in Jeju Strait, Korea. *Front. Mar.*
*Sci.* **9**: 985756. 10.3389/fmars.2022.985756 (2022).

Lenoble V, *et al.* Bioaccumulation of trace metals in the plastisphere:
Awareness of environmental risk from a European perspective. *Enviro.*
*Pollut.* **348**: 123808. 10.1016/j.envpol.2024.123808 (2024).

Lobelle, D & M Cunliffe. Early Microbial Biofilm Formation on Marine Plastic
Debris. *Mar. Pollut. Bull.* **62**, 197–200. 0.1016/j.marpolbul.2010.10.013
(2011).

Lobelle D, *et al.* Global Modeled Sinking Characteristics of Biofouled
Microplastic. *J. Geophys. Res. Oceans.* **126**: e2020JC017098.
10.1029/2020jc017098 (2021).

Luria CM, LA Amaral-Zettler, HW Ducklow HW, JJ Rich. Seasonal
Succession of Free-Living Bacterial Communities in Coastal Waters of the
Western Antarctic Peninsula. *Front. Microbiol.* **7**:1731.
10.3389/fmicb.2016.01731 (2016).

Mendes, CRB, *et al.* Cryptophytes: An emerging algal group in the rapidly
changing Antarctic Peninsula marine environments. *Glob. Change Biol.* **29**,
1791–1808. [10.1111/gcb.16602](https://doi.org/10.1111/gcb.16602) (2023).

Monteiro T, *et al.* Spatiotemporal variability of dissolved inorganic
macronutrients along the northern Antarctic Peninsula (1996–2019). *Limnol.*
*Oceanogr.* **68**: 2305–2326. [10.1002/lno.12424](https://doi.org/10.1002/lno.12424) (2023).

Nguyen D, M Masasa, O Ovadia, L Guttman. Ecological insights into the
resilience of the marine plastisphere throughout a storm disturbance. *Sci.*
*Total Environ.* **858**: 159775. [10.1016/j.scitotenv.2022.159775](https://doi.org/10.1016/j.scitotenv.2022.159775) (2023).

Oberbeckmann S & M Labrenz. Marine Microbial Assemblages on
Microplastics: Diversity, Adaptation, and Role in Degradation. *Ann. Rev.*
*Mar. Sci.* **12**, 209–232. [10.1146/annurev-marine-010419-010633](https://doi.org/10.1146/annurev-marine-010419-010633) (2020).

Oksanen J, *et al.* Package ‘vegan’ Title Community Ecology Package.
*Comm. Ecol. Pack.* **2**(9). [https://cran.r-](https://cran.r-project.org/web/packages/vegan/vegan.pdf)
[project.org/web/packages/vegan/vegan.pdf](https://cran.r-project.org/web/packages/vegan/vegan.pdf) (2019).

Payne MR. Fur Seals *Arctocephalus tropicalis* and *A. gazella* Crossing the
Antarctic Convergence at South Georgia. *Mammalia* **43**, 93–98 (1979).

Pinnell LJ, JW Turner. Temporal changes in water temperature and salinity
drive the formation of a reversible plastic-specific microbial community.
*FEMS Microbiol. Ecol.*, **96**(12), p.fiaa230. [10.1093/femsec/fiaa230](https://doi.org/10.1093/femsec/fiaa230) (2020).

Quast C, *et al.* The SILVA Ribosomal RNA Gene Database Project:
Improved Data Processing and Web-Based Tools. *Nucleic Acids Res.* **41**,
590–596. [10.1093/nar/gks1219](https://doi.org/10.1093/nar/gks1219) (2013).

R Core Team. R: A language and environment for statistical computing. R
Foundation for Statistical Computing. [https://www.R-proje ct.org/](https://www.R-project.org/) (2021).

Ramsperger AFRM, *et al.* Structural Diversity in Early-Stage Biofilm
Formation on Microplastics Depends on Environmental Medium and
Polymer Properties. *Water* **12**(11): 3216. <https://doi.org/10.3390/w12113216>
(2020).

Reisoglu Ş, C Cati, M Yurtsever, S Aydin. Evaluation of prokaryotic and
eukaryotic microbial communities on microplastic-associated biofilms in
marine and freshwater environments. *Eng. Life Sci.* **24**: 2300249.
10.1002/elsc.202300249 (2024).

Reisser J, *et al.* Marine Plastic Pollution in Waters around Australia:
Characteristics, Concentrations, and Pathways. *PLoS ONE* **8**(11): e80466.
10.1371/journal.pone.0080466 (2013).

Rillig, M.C., Kim, S.W. & Zhu, YG. The soil plastisphere. *Nat Rev*
*Microbiol* **22**, 64–74. [10.1038/s41579-023-00967-2](https://doi.org/10.1038/s41579-023-00967-2) (2024).

Rummel CD, A Jahnke, E Gorokhova, D Kühnel, M Schmitt-Jansen. Impacts
of Biofilm Formation on the Fate and Potential Effects of Microplastic in the
Aquatic Environment. *Environ. Sci. Tech. Lett.* **4**(7), 258–267.
10.1021/acs.estlett.7b00164 (2017).

Ryan PG, PJN Bruyn, MN Bester. Regional Differences in Plastic Ingestion
among Southern Ocean Fur Seals and Albatrosses. *Mar. Pollut. Bull.* **104**,
207–210. 10.1016/j.marpolbul.2016.01.032 (2016).

Santos A, *et al.* Measuring the effect of climate change in Antarctic microbial
communities: toward novel experimental approaches. *Curr. Opin.*
*Biotechnol.* **81**: 102918. 10.1016/j.copbio.2023.102918 (2023).

Sangrà P, *et al.* The Bransfield current system. *Deep-Sea Res. I: Oceanogr.*
*Res. Pap.* **58**(4): 390. 10.1016/j.dsr.2011.01.011 (2011).

Sérvulo T, *et al.* Plastisphere composition in a subtropical estuary: Influence
of season, incubation time and polymer type on plastic biofouling. *Environ.*
*Pollut.* **332**: 121873. 10.1016/j.envpol.2023.121873 (2023).

Signori CN, F Thomas, A Enrich-Prast, RC Pollery, SM Sievert. Microbial
diversity and community structure across environmental gradients in
Bransfield Strait, Western Antarctic Peninsula. *Front. Microbiol.* **5**: 647.
10.3389/fmicb.2014.00647 (2014).

Soto DF, I Gómez, P Huovinen. Antarctic snow algae: unraveling the
processes underlying microbial community assembly during blooms
formation. *Microbiome* **11**: 200. 10.1186/s40168-023-01643-6 (2023).

Stoeck T, *et al.* Multiple Marker Parallel Tag Environmental DNA Sequencing
Reveals a Highly Complex Eukaryotic Community in Marine Anoxic Water.
*Mol. Ecol.* **19**: 21–31. 10.1111/j.1365-294X.2009.04480.x (2010).

Suaria G, *et al.* Floating Macro- and Microplastics around the Southern
Ocean: Results from the Antarctic Circumnavigation Expedition. *Environ. Int.*
**136**: 105494; 10.1016/j.envint.2020.105494 (2020).

Sun Y, *et al.* Plastisphere microbiome: Methodology, diversity, and
functionality. *iMeta* **2**: e101. 10.1002/imt2.101 (2023).

Testa G, A Piñones, LR Castro. Physical and Biogeochemical
Regionalization of the Southern Ocean and the CCAMLR Zone. *Front. Mar.*
*Sci.* **8**: 592378. 10.3389/fmars.2021.592378 (2021).

Tuuri EM & SC Leterme. How plastic debris and associated chemicals
impact the marine food web: A review. *Environ. Pollut.* **321**: 21156.
10.1016/j.envpol.2023.121156 (2023).

Van Cauwenberghe L, A Vanreusel, J Mees, CR Janssen. Microplastic
Pollution in Deep-Sea Sediments. *Environ. Pollut.* **182**, 495–499.
10.1016/j.envpol.2013.08.013 (2013)

van Franeker JA & PJ Bell. Plastic Ingestion by Petrels Breeding in
Antarctica. *Mar. Pollut. Bull.* **19**, 672–74. 10.1016/0025-326X(88)90388-8
(1988).

Waller CL, *et al.* Microplastics in the Antarctic Marine System: An Emerging
Area of Research. *Sci. Total Environ.* **598**, 220–227.
10.1016/j.scitotenv.2017.03.283 (2017).

Walters W, *et al.* Improved Bacterial 16S rRNA Gene (V4 and V4-5) and
Fungal Internal Transcribed Spacer Marker Gene Primers for Microbial
Community Surveys. *MSystems.* **1**. 10.1128/mSystems.00009-15 (2016).

Waluda CM & IJ Staniland. Entanglement of Antarctic Fur Seals at Bird
Island, South Georgia. *Mar. Pollut. Bull.* **74**, 244–252.
10.1016/j.marpolbul.2013.06.050 (2013).

Waluda CM, *et al.* Thirty Years of Marine Debris in the Southern Ocean:
Annual Surveys of Two Island Shores in the Scotia Sea. *Environ. Int.* **136**:
105460. 10.1016/j.envint.2020.105460 (2020).

Wickham H. Ggplot2 by Hadley Wickham. *Media* **35**: 211. 10.1007/978-0-
387-98141-3 (2009).

Wright RJ, G Erni-Cassola, V Zadjelovic, M Latva, JA Christie-Oleza. Marine
plastic debris: A new surface for microbial colonization. *Environ. Sci.*
*Technol.* **54**, 11657–11672. 10.1021/acs.est.0c02305 (2020).

Wright RJ, MGI Langille, TR Walker. Food or just a free ride? A meta-
analysis reveals the global diversity of the plastisphere. *The ISME Journal*
**15**(3), 789–806. [10.1038/s41396-020-00814-9](https://doi.org/10.1038/s41396-020-00814-9) (2021).

Zettler ER., TJ Mincer, L Amaral-Zettler. Life in the ‘Plastisphere’: Microbial
Communities on Plastic MarineDebris. *Environ. Sci. Technol.* **47**, 7137-7146
(2013).